# Should We Really Edit Language Models?
# On the Evaluation of Edited Language Models

**Qi Li**$^{\heartsuit*}$, **Xiang Liu**$^{\heartsuit*}$, **Zhenheng Tang**$^{\clubsuit}$, **Peijie Dong**$^{\heartsuit}$,
**Zeyu Li**$^{\heartsuit}$, **Xinglin Pan**$^{\heartsuit}$, **Xiaowen Chu**$^{\heartsuit\dagger}$

$\heartsuit$ The Hong Kong University of Science and Technology (Guangzhou)
$\clubsuit$ Hong Kong Baptist University
lqinfdim@163.com   xliu886@connect.hkust-gz.edu.cn
{pdong212, zli755, xpan413}@connect.hkust-gz.edu.cn
zhtang@comp.hkbu.edu.hk   xwchu@hkust-gz.edu.cn

## Abstract

Model editing has become an increasingly popular method for efficiently updating knowledge within language models. Current approaches primarily focus on reliability, generalization, and locality, with many excelling across these criteria. Some recent studies have highlighted the potential pitfalls of these editing methods, such as knowledge distortion and conflicts. However, the general capabilities of post-edited language models remain largely *unexplored*. In this paper, we conduct a comprehensive evaluation of various editing methods across different language models, and have the following findings. (1) Existing editing methods inevitably lead to performance deterioration on general benchmarks, indicating that these methods can maintain the general capabilities of the model only within a limited number of edits. When the number of edits increases, the model's intrinsic knowledge structure may be disrupted or even completely destroyed. (2) Instruction-tuned models exhibit greater robustness to editing, showing less performance drop on general knowledge after editing. (3) Language models with larger scale are more resistant to editing compared to smaller models. (4) The safety of the edited models is significantly compromised, even for models that were originally safety-aligned. Our findings indicate that current editing methods are only suitable for small-scale knowledge updates within language models, which motivates further research on more practical and reliable editing methods. The details of code and reproduction can be found in Appendix F.

## 1  Introduction

Recently, large language model (LLM) like ChatGPT [1], Claude [2], and Llama [3; 4] have revolutionized the deep learning and demonstrated remarkable performance across various knowledge-intensive tasks [5; 6; 7]. However, the learned vast amount of knowledge in LLMs may be erroneous, harmful, or outdated [8]. While directly fine-tuning an LLM on calibrated knowledge can help mitigate this problem, which is prohibitive due to hardware constraints and resource budget [9; 10; 11]. To this end, *model editing* [12; 13; 14] has been proposed to efficiently update knowledge within LLM. Current studies in editing [15; 16; 17; 18] target at enabling efficient yet precise model behavior alterations on specific knowledge samples in the form of triplet within LLMs like modifying the wrong tuple (Tim Cook, is the CEO of, Google) to the correct one (Tim Cook, is the CEO of, Apple) persistently.

---

*Equal Contribution.
$\dagger$Corresponding Author.

38th Conference on Neural Information Processing Systems (NeurIPS 2024).

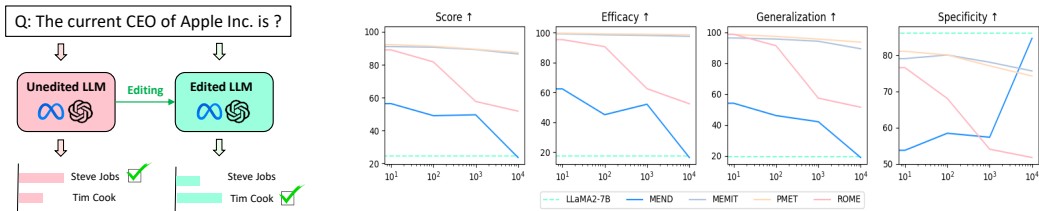

Figure 1: Illustration about the model editing and its pitfalls in retaining edited knowledge. Left panel: model editing methods can efficiently update knowledge within language models; Right panel: when scaling editing to thousands, the model can't retain edited knowledge, see [17] for details.

The primary goal of model editing is to impact the predictions for related inputs termed editing scope [19] generally, *without* influencing behaviors on unrelated knowledge samples. The assessment of an editing method typically involves evaluation along three dimensions [14; 12]. First and foremost, the *reliability* aims to ascertain the capability of the post-edited model to recall the specific editing fact accurately. Second, the *generalization* seeks to validate the adaptability of the edited model by assessing its ability to recall the editing fact under diverse paraphrase prompts. The last dimension *locality* (a.k.a., specificity) is employed to verify the robustness of the edited model by examining whether its output for unrelated inputs remains consistent after editing. Existing knowledge editing methods like SERAC [18], ROME [17], MEMIT [20], and IKE [15] work well on these evaluation criteria across various datasets on different LLMs.

Despite these successes in editing language models, recent works [21; 22; 23; 24] have disclosed the inevitable pitfalls of existing editing methods from different perspectives such as knowledge distortion [23], and catastrophic forgetting [25]. In sequential editing setting (see Section 2), as the number of edits increases, it is necessary to balance two aspects: the retention of the model's original knowledge and the preservation of newly acquired knowledge through updates. These two objectives are to some extent conflicting. The *general capabilities* (Section 2) of LLMs is the foundation to solve the wide range of complex tasks. Changes in the model's general capabilities reflect the retention of its original knowledge. However, the general capabilities of post-edit language models are still *unexplored*, making current editing methods unreliable to be employed for real-world applications. Given this situation, it naturally motivates the following critical question to explore:

*How do sequential model editing affect the general abilities of language models?*

To close this gap, we make a comprehensive understanding and analysis of edited LLMs with various editing methods (In Section 3). In detail, we edit multiple LLMs with various editing methods and evaluate them across benchmarks to verify underlying factors that may affect the general abilities. It is worth noting that our focus is on the *general capabilities* of the model (including world knowledge, reading comprehension, reasoning, safety, etc.), rather than the performance on *efficacy*, *generalization* and *locality* or downstream tasks like NER, QA, and NLI. These distinctions distinguish our work from some existing studies like [26; 25; 27]. Technically, we explore the impact of various underlying factors such as the number of edits, model scale, safety, different aspects of abilities, and instruction tuning on the general capabilities of edited LLMs after sequential editing.

The empirical results indicate that the majority of current editing methods do not significantly influence the fundamental capabilities of models within dozens of edits (Section 4.1). However, after nearly to hundred edits, some methods lead to rapid degradation, while other methods only slightly affect performance after several hundred edits or even thousands (MEND, PMET). When subjecting models to an extremely high number of edits (up to $10k$), we observe that the intrinsic knowledge structure of the models is thoroughly disrupted, leading to outputs of empty strings for any input. We refer to this phenomenon as the *muting effect* in sequential language model editing.

Furthermore, we discovered that for the vast majority of current editing methods, the instruction-tuned models exhibit a slower rate of performance decline after edits (Section 4.2), and the smaller model is more vulnerable to deterioration caused by editing (Section 4.3). Moreover, results reveal that different model editing methods affect all aspects of a model's capabilities to a roughly equivalent extent (Section 4.4). Our research also explores the safety aspects of these edited models (Section 4.5), revealing that even with dozens of edits, safety can be compromised to a certain extent.

We hope our work can provide the research community with insightful perspectives to help advance studies in this field, systematically elucidating the impact of existing model editing methods on LLM. To the best of our knowledge, we are the *first* to comprehensively evaluate and analyze the general capabilities of edited language models. Our main contributions are summarized as follows:

- We conducted a detailed evaluation of the impact of different model editing methods on the general capabilities of LLM across various numbers of edits. We found that existing model editing methods are only suitable for a limited number of edits, generally not exceeding several dozens.

- Empirically, extensive explorations with different editing approaches were conducted to verify potential factors affecting the fundamental capabilities of edited models. These insights are broadly applicable across different editing methods and various models.

- Our empirical studies with various editing methods across different models reveal that even with only dozens of edits, the safety of LLMs can be compromised to a certain extent.

- Technically, we conducted an in-depth analysis of the side effects, operational efficiency, and deployment of edited LLM, and discussed their practical use in production.

## 2 Preliminary

In this section, we provide comprehensive preliminaries of model editing and LLM evaluation.

**Model Editing.** Model editing (also known as knowledge editing) aims to precisely adjust the behaviors of a language model $M_\theta$ on some facts without influencing unrelated samples. Current works focus on editing knowledge tuple $t = (s, r, o)$. The editing process inserts new tuples $(s, r, o^*)$ in place of the current tuple $(s, r, o)$, where these two share the same $s$ and $r$. An editing operation is denoted as $e = (s, r, o, o^*)$ for brevity. Given $n$ fact tuples $T^* = (t_1^*, t_2^*, \ldots, t_n^*)$ where $t_i^* = (s_i, r_i, o_i^*), i = 1, 2, \ldots, n$, and a model $M_\theta$, model editing yields an edited language model $M_\theta^e$ via editing operations $\mathcal{E} = \{e_1, e_2, \ldots\}$, where $M_\theta^e(s_j, r_j) = o_j^*$ if $t_j = (s_j, r_j, o_j^*) \in T^*$, else $M_\theta^e(s_j, r_j) = o_j$. To evaluate model editing methods, current works focus on three dimensions: *reliability*, *generalization*, and *locality* [19]. Please refer to Section 6 for a comprehensive survey.

**General Capabilities of Language Models.** In recent years, the field of LLM has experienced rapid growth, leading to the development of numerous models by various research institutions. These models differ significantly in terms of parameter size, architecture, corpora, and training methodologies. Consequently, it has become critically important to evaluate the capabilities of these models objectively, and comprehensively. Typically, this is achieved by evaluating the models on widely adopted benchmarks like MMLU [28], and BigBench [29] to compare their performance with their counterparts. Currently, the evaluation of the general capabilities of LLMs in both academia and industry focuses on several key areas: world knowledge, common sense reasoning, coding, reading comprehension, mathematical skills, and performance on mainstream benchmark datasets [30; 3; 4; 31; 32; 33]. This paper concentrates on the impact of editing on the inherent capabilities of LLM, rather than on downstream tasks like QA, NER, and NLI in work [26].

**Model Editing Evaluation.** The current evaluation protocol of model editing involves updating a model's knowledge, assessing the post-edit model, and restoring the update before editing. However, in real-world applications like *sequential editing*, models are expected to maintain preceding modifications before performing new edits. Scaling sequential editing capability is therefore crucial for model editing. In this paper, we mainly focus on whether the general abilities of LLM are hurt in sequential editing settings. Two orthogonal concepts of sequential editing are *single editing* and *batch editing*. Batch editing refers to the model's ability to edit multiple editing samples (like MEMIT, PMET) at once, whereas the concept opposed to this is single editing, which means these methods (like ROME, GRACE, MEND) can only edit one sample at a time.

**Formal Definition of Sequential Editing.** Here, let's provide a formal definition of sequential editing. Assume we have an unedited model $M_0$, and $n$ editing samples $(x_i, y_i)$, where $i = 1, 2, \ldots, n$ need to be incorporated into the language model $M_0$. Suppose the editing operation is a function $E(\cdot, \cdot)$, where the first parameter is the model to be edited and the second parameter is the editing samples. Assume we get the edited model $M_i$ after the $i$-th editing operation. In sequential editing, $M_t$ (model parameter after the $t$-th editing) is determined by the model weight $M_{t-1}$ and the editing sample used in the $t$-th edit, like $M_t = E(M_{t-1}, S_t)$, where $S_t$ is the edit samples used in the $t$-th

edit. For different index $i$ and $j$, $S_i \cap S_j = \emptyset$; for every $i$ , we have $\bigcup_i S_i = \{x_j, y_j\}_{j=1}^{j=n}$. If we denote the size of $S_t$ is $n_t$, it satify $n_t \geq 1$ for every $t$ and satisfies $n = \sum_t n_t$ for all edit batches.

# 3 Experimental Design

In this section, we present several critical parts of experimental setups (in Section 3.1) and research question designs (in Section 3.2). The results and analysis of research questions are left in Section 4. All of the details of reproducing our experiments can be found in Appendix F.

## 3.1 Experimental Setups

Here, we list all of the experimental setups. For implementation details, please refer to Appendix C.

**Language Models.** We conduct experiments on various LLMs, including Llama2-7B (based and instruction-tuned) [4], Mistral-7B (based and instruction-tuned) [31], GPT2-XL [34], and 6 language models from Pythia [35] model families with varying parameter scale from 160M to 12B.

**Model Editing Methods.** To comprehensively investigate the potential impact on edited models, we compared multiple editing methods: (1) meta-learning based methods: MEND [16] (2) located-then-edit based method: KN [36], ROME [17], MEMIT [20], PMET [37], (3) retrival based methods: SERAC [18], (4) extra parameters based methods: GRACE [38]. It is imperative to note that the efficacy of edited models varies depending on the hyperparameters employed for each method, which can significantly impact the performance of the edited model. Therefore, our evaluation was confined to assessing only models supported by each editing method.

**Editing Datasets.** In this work, we employ the most widely adopted editing datasets **ZsRE** [39] and COUNTERFACT [17] as editing datasets across all experiments in the work.

**Evaluation Benchmarks.** To effectively determine whether the model editing influences the overarching capabilities of LLMs, we utilized five distinct task categories as benchmarks. These include: *World Knowledge:* MMLU [28] (5-shot), BBH [29] (3-shot), with the assessment based on the accuracy of multiple-choice answers. *Arithmetic:* GSM8K [40] (8-shot), evaluated by the solve rate. *Commonsense Reasoning:* CommonsenseQA [41] (7-shot), where performance is gauged by the accuracy of multiple choices. *Reading Comprehension:* TriviaQA [42] (0-shot), with the assessment based on the exact match from context. *Safety*: TruthfulQA [43] (0-shot), evaluated through multiple-choice accuracy, and ToxiGen [44] (0-shot), where results are determined by the accuracy of two-way classification. For more details of these benchmarks, please refer to Appendix D.1.

**Editing Settings.** To investigate the potential impact on language models caused by editing, we mainly focus on sequential single editing in most of the research questions in this work.

## 3.2 Research Questions

This paper aims to thoroughly examine the impacts of diverse model editing methods on various general abilities of edited models. It naturally motivates the following critical research questions (RQs) to be explored in this work based on the primary aim.

- **RQ1:** How does the number of undergone edits affect the abilities of models? (In Section 4.1)
- **RQ2:** Do instruction-tuned models exhibit differently than base counterparts? (In Section 4.2)
- **RQ3:** Does the general abilities of the edited model differ on model scales? (In Section 4.3)
- **RQ4:** How does editing affects different aspects of a model's capabilities? (In Section 4.4)
- **RQ5:** Does performing editing on language models compromise their safety? (In Section 4.5)

In the next section, we will address these research questions through detailed experimentation.

# 4 Results and Analysis

In this section, we present empirical results and a comprehensive analysis of research questions. More detailed information and conducted experiments are presented in Appendix C. The case studies of benchmark evaluation with different editing settings can be found in Appendix A.

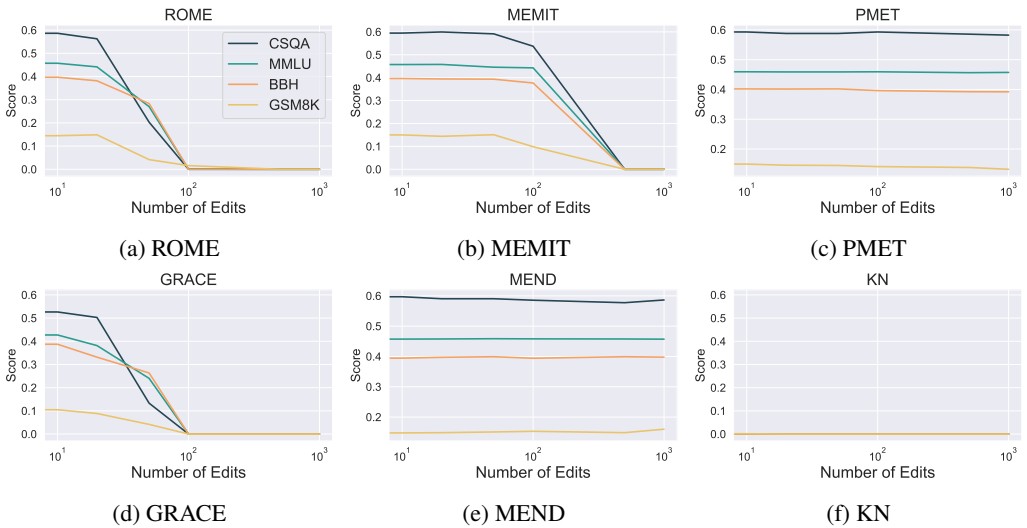

Figure 2: Performance trends of evaluating edited Llama2-7B base model across different benchmarks using six editing methods. Results reveal that PMET and MEND can effectively preserve the model's abilities across all tasks. While KN drastically drops even less than ten edits.

## 4.1 RQ1: Impact of the Number of Edits

**Main results.** We first investigate the impact of the number of edits with different methods (RQ1) on the general abilities of edited language models here. We perform editing with various editing methods on Llama2-7B, and Mistral-7B with different numbers of edits. The detailed results are reported in Figure 2 and Table 2. The results reveal that models edited with various methods exhibit significant performance divergences after undergoing different numbers of edits. Experiments on the Llama2-7B model demonstrate that the majority of model editing techniques maintain the original capabilities of the model with fewer than 20 edits. However, when the number of edits increases, some methods, such as ROME and MEND, exhibit significant performance degradation. In contrast, other approaches, like PMET, do not affect model performance even after hundreds of edits.

**The muting effect: scaling sequential single editing to $10k$ edits.** We have demonstrated that editing methods such as MEMIT and PMET can enable models to withstand hundreds of edits while maintaining their original capabilities as much as possible. However, the extent to which the number of edits can be expanded remains an *open question*. To investigate the performance of language models after undergoing an extremely large number of sequential edits, we applied the ROME, MEMIT, and PEMT methods to the Llama, Mixtal, and GPT2-XL on the COUNTERFACT dataset, implementing 10,000 sequential edits. The results revealed that after such a substantial number of edits, the intrinsic knowledge structure of the models was completely disrupted. For any input, the response was an **empty string** or **random character**. We refer to this phenomenon as the ***muting effect*** of model editing. Please refer to the Case Study section, Section A, for more details.

**Finding 4.1.** The majority of existing methods can only undergo dozens of edits without compromising performance, while only a few methods can scale to thousands of edits.

## 4.2 RQ2: Does Instruction Tuned LLM Show Better Performance after Editing?

We then explore the impact of instruction tuning (RQ2) here. The empirical results are demonstrated in Figure 3 and Table 3. Our findings indicate that for the majority of editing approaches, the impact on performance after editing is comparable between the instruction-tuned model and the base model. However, instruction tuning can slow down the rate of performance decline after model edits. Performance is not significantly affected by lower than 20 edits. However, some methods exhibit noticeable performance degradation after exceeding 20 edits. Notably, when the MEMIT method is used to edit models that have been fine-tuned with instructions, there is a slower decline in general capabilities with increasing edits compared to models that have not undergone instruction tuning.

**Finding 4.2.** Instruction-tuned model exhibits a slower rate of performance decline after editing.

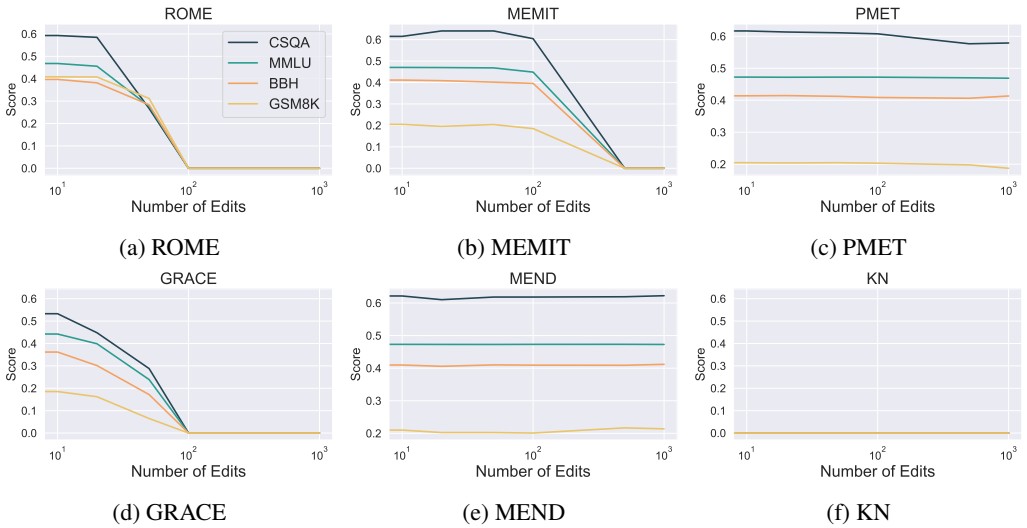

Figure 3: Performance trends of assessing edited Llama2-chat-7B across different benchmarks using 6 editing methods. Results reveal that PMET and MEND can effectively preserve the model's abilities across all tasks. While KN drastically drops even less than ten edits.

## 4.3 RQ3: Do the General Abilities of the Edited Model Differ on Model Scales?

In this subsection, we examine the impact of model size (RQ3) on the capabilities of edited models. To ensure a fair comparison, we employ multiple models from the Pythia model family, with sizes ranging from 160M to 12B parameters. We perform editing with two methods: MEMIT and ROME, and then conduct evaluation on 4 benchmarks. We summarize the empirical results of general abilities evaluation in Figure 4 and Table 4 with different model sizes. Our findings reveal that for some methods, such as ROME, the rate of performance degradation in language models following edits slows as the model size increases. Conversely, the impact of other editing methods, like MEMIT and PMET, on post-edit language model general abilities appears to be independent of model parameters.

**Finding 4.3.** Larger models exhibit less side effect on benchmarks after editing.

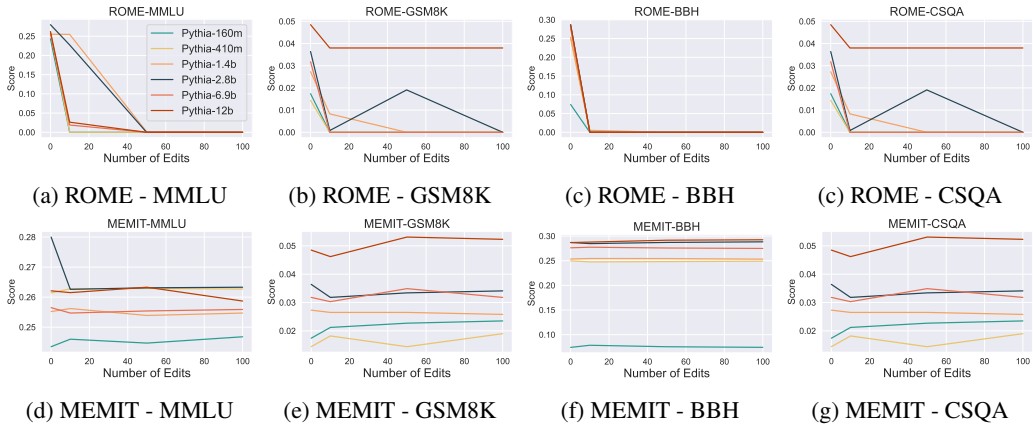

Figure 4: Quantitative results of exploring the impact of model scale on edited language models. We perform editing with different-size models in Pythia model families with ROME (the first row) and MEMIT (the second row), and then these models are evaluated across diverse benchmarks.

### 4.4 RQ4: How Does Editing Affect Different Aspects of a Model's Capabilities?

In this subsection, we would like to further explore how model editing influences different aspects of the edited language model abilities (RQ4). Figure 5 comparing 6 different editing methods on the Llama2-7B model across various benchmarks. Different benchmarks correspond to different abilities of the language model, e.g. MMLU and BBH are for world knowledge, GSM8K corresponds to math abilities, TrivialQA is for reading comprehension, and CSQA is related to reasoning. Results in Table 2 and 3 reveal that different model editing methods affect all aspects of a model's capabilities roughly uniformly. Results in Table 2 and 3 reveal that different model editing methods affect all aspects of a model's capabilities uniformly. Figure 5 comparing 6 different editing methods on the Llama2-7B model across various benchmarks reveals that PMET and MEND maintain the most consistent performance, effectively preserving the model's abilities across all tasks, even with numerous edits.

**Finding 4.4.**    Editing affects the different capabilities of LLM to a roughly equivalent extent.

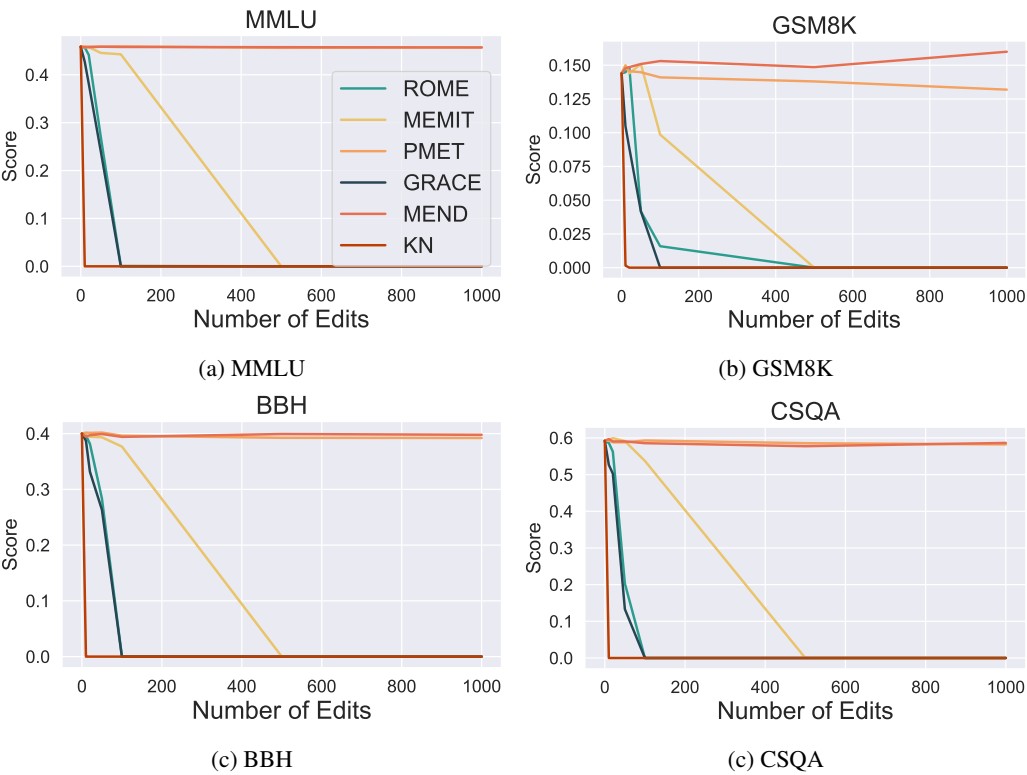

(a) MMLU    (b) GSM8K

(c) BBH    (c) CSQA

Figure 5: Evaluation of different kinds of general capabilities of edited language models. Results reveal that PMET and MEND can effectively preserve the model's abilities across all tasks.

### 4.5 RQ5: The Safety Cost of Editing Language Models

We inspect the safety issues caused by performing editing on LLM (RQ5) in this subsection. To adjust pre-trained LLMs' behavior on specific cases, some editing operations are desirable. But, what are the safety costs associated with these editing operations? Here, we focus on studying the ability of edited LLMs to handle malicious prompt attacks and adversarial input. We conducted evaluations on the ToxiGen dataset and TruthfulQA with LLM that edited with various methods and different edits. In Figure 6 and Table 5, we summarize the experimental results of safety evaluation on edited language models with different methods. These experimental results indicate that the security of edited LLMs (including alignment with human values, resilience to malicious inputs, and prevention of generating harmful content) is compromised to some extent as the number of edits increases, even with instruction tuning. Interestingly, many methods exhibit improved scores when the model has undergone 100 edits. We hypothesize that this is due to the significant disruption of the model's intrinsic knowledge structure.

**Finding 4.5.** Even dozens of edits can compromise the safety of edited language models.

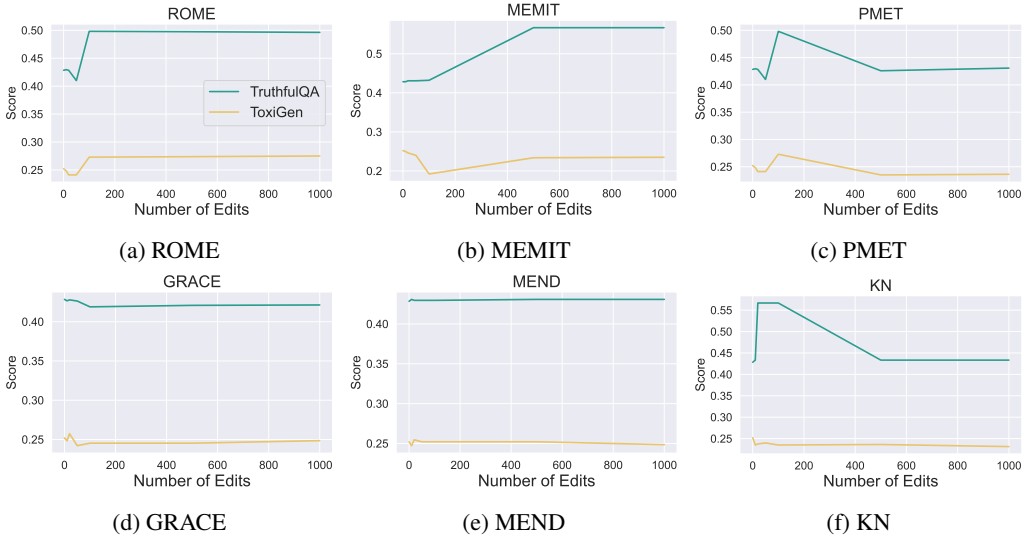

(a) ROME      (b) MEMIT      (c) PMET

(d) GRACE      (e) MEND      (f) KN

Figure 6: Safety evaluation of edited language models. We perform evaluation on TruthfulQA and Toxigen datasets with the Llama2-7B model. Results demonstrate that for most editing methods, even dozens of edits can compromise the safety of language models with they are aligned.

**Summary.** From the above research questions, we conclude that existing editing methods have inevitable pitfalls in editing LLMs, making them impractical in the production environment.

## 5 Further Discussion

In this section, we conduct discussions on the side effects, editing efficiency, and deployment issues of post-edit LLM, to further explore their practical use in production environments.

**Potential Impact on Inherent Knowledge within LLM.** Existing model editing methods claim that they can update specific knowledge within LLM without affecting other unrelated knowledge. However, existing research [26; 25] indicates that methods involving direct modification of model parameters can have unintended and potentially harmful impacts on the intrinsic knowledge of the model. Moreover, the updated knowledge is often challenging to utilize in knowledge-intensive downstream tasks, such as reasoning and knowledge-based question answering, exacerbating the model's hallucinations [22; 21]. As the quantity of edits escalates, the retention of updated knowledge within the LLM markedly diminishes [20]. Our series of experiments demonstrates that even with only hundreds of edits, the general capabilities of the model are severely compromised. When the number of edits reaches the thousands, the model's internal structure is thoroughly damaged. These findings highlight the deficiencies of current model editing methods.

**Editing Efficiency and Speed.** Existing editing methods claim to manipulate the inherent knowledge within model efficiently and rapidly while keeping the memory usage of LLMs close to that of vanilla model inference during deployment. These advantages make model editing seem highly appealing. However, inherent drawbacks severely impede their real-world applicability. Many existing editing methods heavily rely on hyperparameter settings, some even necessitating costly parameter searches. For instance, locate-then-edit approaches (see Section 6) like ROME [17], and PMET [37] require causal tracing to identify layers for editing. Moreover, certain editing methods entail pre-training (MEND [16], SERAC [18], KN [36]) or precomputation of intermediate cached key-value (MEMIT[20]), significantly prolonging pre-training or computational time and consuming hardware resources far beyond those required during editing, without targeted optimization for parallelization benefits. For example, utilizing the MEMIT for $10K$ edits on Llama2-7B [4] with an RTX A6000 48G RAM GPU demands approximately 120 hours. Hence, enhancing the efficiency of existing methods becomes paramount for real-world applications. See Table 8 for edit speed.

**Deployment and Serving.** As the scaling up of LLM, considerable deployment and serving challenges are presented [45; 10; 11]. To reduce the expenses of deployment and serving of LLMs, several specialized frameworks have been developed, including TensorRT-LLM [46], vLLM [47], and LightLLM [48], with high optimization for efficient LLM serving and deploying. However, some model editing methods, such as the GRACE [38], introduce additional modules to the edited models, or employ auxiliary models, like the SERAC [18], to handle queries involving updated knowledge. These modifications to the model architecture prevent the edited models from being directly deployed using these aforementioned serving frameworks, significantly limiting the practical adoption of editing methods. Furthermore, as the number of edits on LLM increases, for methods that introduce additional, the memory consumption for storing and the time cost for retrieval become increasingly prohibitive. This significantly impacts the throughput and inference latency during deployment and serving. For further details on these effects, please refer to Appendix D.2.

## 6 Related Work

In this section, we briefly review the related works in model editing and language model evaluation.

**Model Editing.** Model editing aims to precisely modify knowledge within a language model with fine grain. Existing methods can be divided into 3 different categories: Retrieval-based, Extra-parameters-based, and Locate-then-edit-based methods. Early works on editing focused on updating individual neurons using constrained fine-tuning [49; 50; 51], or hypernetworks [52]. A related line of work has focused on storing updates in an external memory [53]. Inspired by the linear associative memory property of FFN in transformers [54] and success with the approach in convolutional models [55], recent Locate-then-edit style works have proposed to edit MLP weights directly [17; 20; 37]. In the encyclopedic factual domain, work [17] proposed to edit single facts by fitting a Rank One Model Edit (ROME) to the parameters of an MLP layer and showed that it outperformed prior methods. While [56] concentrates on editing commonsense knowledge. Work [57] focuses on editing the encoder-decoder model. Work [12; 14; 19] make a comprehensive survey in model editing.

**Pitfalls of Model Editing.** Since the concept of model editing was introduced, there exist few works to discuss the drawbacks of existing methods. In [23], two kinds of pitfalls are discovered, named knowledge conflict and knowledge distortion. MEMIT-CSK [56] finds common sense knowledge can also be localized in FFN of LLM layers and both subject, object, and relation play important roles in recalling memory, but they only focus on classification tasks. [58] focus on mitigating the reversal curse in model editing for LLM. Some recent works [22; 59] try to evaluate the multi-hop reasoning of updated knowledge. Work [27; 21] tries to assess the underly impact caused by editing. What's more, some works [60; 61; 62; 63] discuss the knowledge forgetting during fine-tuning.

**Evaluation of LLM.** Evaluating the effectiveness of LLMs [3; 4; 31; 30; 64; 32; 33] involves a diverse array of tests, where models are assessed across various tasks, showcasing their capabilities. Among the benchmarks employed, Bigbench [65; 29], MMLU [28], and HELM [66] are notable. These benchmarks utilize a range of automatic evaluation metrics, such as BLEURT [67] and others requiring meticulous annotation to ensure data quality for downstream applications [68; 69; 70]. Typically, these evaluations emphasize accuracy across multiple choices, which serves as the primary metric [71; 72]. Foundation models, often tested across broad linguistic tasks—both generative and multiple-choice—are analyzed to ensure rigorous assessment standards [73; 28; 74; 65]. These methods, however, may focus on the accuracy within confined options and might not effectively capture the nuances of more open-ended, practical applications where models generate free-form text. Moreover, assessing the safety of LLMs is essential. Safety evaluations concentrate on three key areas: truthfulness [43], toxicity [44], and bias [75].

## 7 Conclusion

In this work, we systematically investigate the potential impact of model editing on language models. We employed various editing methods to modify multiple models, followed by evaluations across different benchmarks. The experimental results indicate that existing editing methods can preserve the general capabilities of the model within a limited number of edits, not exceeding a few dozen. When the number of edits is sufficiently large, the intrinsic knowledge structure of the model can be disrupted or even completely damaged. Additionally, we systematically investigated potential factors

that might impact the performance of models post-editing and their potential effects on the model's basic capabilities. Our experiments demonstrate that after only a few dozen edits, the safety of the model is compromised, including those models that have been aligned. Furthermore, we discuss the side effects, operational efficiency issues, and potential impacts brought by editing operations. We conclude that most of existing editing methods are only suitable for scenarios with no more than a few dozen updates. We are looking forward to future research addressing these challenges.

## Acknowledge

This work was partially supported by National Natural Science Foundation of China under Grant No. 62272122, the Guangzhou Municipal Joint Funding Project with Universities and Enterprises under Grant No. 2024A03J0616, the Hong Kong RIF grant under Grant No. R6021-20, and Hong Kong CRF grants under Grant No. C2004-21G and C7004-22G.

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

## Limitation

In this paper, we explore the potential impact of model editing on the general abilities of language models. We conduct evaluation on various language models across diverse benchmarks to answer research questions. However, there are still some limitations to be further addressed: (1) limited editing methods: we only evaluate 6 editing methods; (2) limited benchmarks: only 4 benchmarks are involved in main experiments ; (3) more deep analysis: further discussion and analysis are needed. We hope to address these problems in sequential works.

## Impact Statement

In this paper, we systematically investigate the general abilities of the edited language model. Our study highlights the potential impact caused by existing editing methods on language models. Given the prominent prospect of editing language models, our approach comprehensively evaluates edited language models. Consequently, this aids in reducing and mitigating the generation of harmful or biased content. On the other hand, editing methods could also be utilized to inject harmful information into open-source language model weights, potentially leading to significant societal impacts.

# Appendix and Supplementary Material

# A  Case Study

In this section, we would like to present a detailed case study of each model editing method. Below instances are different editing methods applied to the Llama2-7B model, the experiment setting is aligned with section 3. The question is randomly picked from the GSM8K dataset, and query with 8 Chain-of-Thought prompts. Figure 7 shows the answer from vanilla Llama2-7B.

> **Question: A new program had 60 downloads in the first month. The number of downloads in the second month was three times as many as the downloads in the first month, but then reduced by 30% in the third month. How many downloads did the program have total over the three months?**

> **Ground Truth:**
> The number of downloads of the program in the second month increased to 3*60 = «3*60=180»180. In the first two months, the total number of downloads of the program was 180+60 = «180+60=240»240. In the third month, the number of downloads of the program reduced by 30/100*180 = «30/100*180=54»54. There were 180-54 = «180-54=126»126 downloads in the third month. In the three months, the total number of downloads of the program was 126+240 = «126+240=366»366

> **Llama2-7B:**
> The program had 60 downloads in the first month. The number of downloads in the second month was 3 times as many as the downloads in the first month. So 60 * 3 = 180 downloads in the second month. The number of downloads in the third month was 30% less than the downloads in the second month. So 180 * 0.7 = 126 downloads in the third month. The answer is 60 + 180 + 126 = 366. ✅

Figure 7: The question and ground truth are from GSM8K, and answer from vanilla Llama2-7B.

## A.1  10 Edits

Figure 8 displays the answers generated by the Llama2-7B model after 10 edits using ROME, MEMIT, PMET, MEND, and KN methods. Observations indicate that after 10 edits using the KN method, the Llama2-7B model fails to produce content resembling human-like language, as its intrinsic knowledge structure was completely disrupted. We refer to this phenomenon as the *muting effect* of model editing. And answers generated by the rest editing methods remain the same as the answer generated by vanilla Llama2-7B.

**Question: A new program had 60 downloads in the first month. The number of downloads in the second month was three times as many as the downloads in the first month, but then reduced by 30% in the third month. How many downloads did the program have total over the three months?**

**ROME:**
The program had 60 downloads in the first month. The number of downloads in the second month was 3 times as many as the downloads in the first month. So 60 * 3 = 180 downloads in the second month. The number of downloads in the third month was 30% less than the downloads in the second month. So 180 * 0.7 = 126 downloads in the third month. The answer is 60 + 180 + 126 = 366. ✅

**MEMIT:**
The program had 60 downloads in the first month. The number of downloads in the second month was 3 times as many as the downloads in the first month. So 60 * 3 = 180 downloads in the second month. The number of downloads in the third month was 30% less than the downloads in the second month. So 180 * 0.7 = 126 downloads in the third month. The answer is 60 + 180 + 126 = 366. ✅

**PMET:**
The program had 60 downloads in the first month. The number of downloads in the second month was 3 times as many as the downloads in the first month. So 60 * 3 = 180 downloads in the second month. The number of downloads in the third month was 30% less than the downloads in the second month. So 180 * 0.7 = 126 downloads in the third month. The answer is 60 + 180 + 126 = 366. ✅

**MEND:**
The program had 60 downloads in the first month. The number of downloads in the second month was 3 times as many as the downloads in the first month. So 60 * 3 = 180 downloads in the second month. The number of downloads in the third month was 30% less than the downloads in the second month. So 180 * 0.7 = 126 downloads in the third month. The answer is 60 + 180 + 126 = 366. ✅

**KN:**
Cong in the English in the English in the English in the English in the English in the English in C*C *C* and the English in the English in C *C * in the English in the English in C *C * and the English in the English and the English and C *C * and the English and the English and C *C * . . . ❌🤯

Figure 8: The answer generate by Llama2-7B after 10 times editing with different methods.

## A.2 50 Edits

Figure 9 displays the answers generated by the Llama2-7B model after 50 edits using the ROME, MEMIT, PMET, and MEND methods. Observations show that after 50 edits using the ROME and MEMIT methods, the model produced incorrect answers, albeit in a human-readable format. Meanwhile, the answers generated by the PMET and MEND methods remain unchanged from those produced by the unedited Llama2-7B model.

> **Question: A new program had 60 downloads in the first month. The number of downloads in the second month was three times as many as the downloads in the first month, but then reduced by 30% in the third month. How many downloads did the program have total over the three months?**

> **ROME:**
> A new program had 60 downloads in the first month. The number of downloads in the second month was 3 times as many as the downloads in the first month. But then in the third month, the number of downloads was reduced by 30%. So 3 * 3 = 9. The answer is 9. ✗

> **MEMIT:**
> The program had 60 downloads in the first month. The number of downloads in the second month was 3 times as many as the downloads in the first month. So 60 * 3 = 180 downloads in the second month. The number of downloads in the third month was 180 - 30% = 126 downloads. So 180 + 126 = 306 downloads in the third month. The answer is 306. ✗

> **PMET:**
> The program had 60 downloads in the first month. The number of downloads in the second month was 3 times as many as the downloads in the first month. So 60 * 3 = 180 downloads in the second month. The number of downloads in the third month was 30% less than the downloads in the second month. So 180 * 0.7 = 126 downloads in the third month. The answer is 60 + 180 + 126 = 366. ✅

> **MEND:**
> The program had 60 downloads in the first month. The number of downloads in the second month was 3 times as many as the downloads in the first month. So 60 * 3 = 180 downloads in the second month. The number of downloads in the third month was 30% less than the downloads in the second month. So 180 * 0.7 = 126 downloads in the third month. The answer is 60 + 180 + 126 = 366. ✅

Figure 9: The answer generate by Llama2-7B after 50 times editing with different methods.

### A.3   100 Edits

Figure 10 displays the answers generated by the Llama2-7B model after 100 edits using the ROME, MEMIT, PMET, and MEND methods. Observations show that after 100 edits using the ROME methods, the *muting effect* phenomenon happened. Meanwhile, the answers generated by the PMET and MEND methods remain unchanged from those produced by the unedited Llama2-7B model.

> **Question: A new program had 60 downloads in the first month. The number of down-loads in the second month was three times as many as the downloads in the first month, but then reduced by 30% in the third month. How many downloads did the program have total over the three months?**

> **ROME:**
> 4. 4. 4. 4. 4. 4. 4. 4. 4.8. 4.8. 4.8. ... 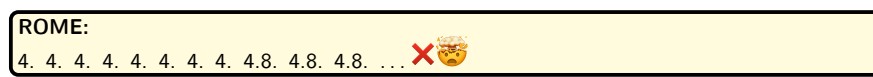

> **MEMIT:**
> The program started with 60 downloads. In the second month, the number of downloads was 3 times as many as the downloads in the first month. So 60 * 3 = 180 downloads. In the third month, the number of downloads was reduced by 30%. So 180 - 30% = 126 downloads. The answer is 126. ✗

> **PMET:**
> The program had 60 downloads in the first month. The number of downloads in the second month was 3 times as many as the downloads in the first month. So 60 * 3 = 180 downloads in the second month. The number of downloads in the third month was 30% less than the downloads in the second month. So 180 * 0.7 = 126 downloads in the third month. The answer is 60 + 180 + 126 = 366. ✅

> **MEND:**
> The program had 60 downloads in the first month. The number of downloads in the second month was 3 times as many as the downloads in the first month. So 60 * 3 = 180 downloads in the second month. The number of downloads in the third month was 30% less than the downloads in the second month. So 180 * 0.7 = 126 downloads in the third month. The answer is 60 + 180 + 126 = 366. ✅

Figure 10: The answer generate by Llama2-7B after 100 times editing with different methods.

# B More Visualization of Results

In this section, we would like to provide more visualization of our experimental results.

## B.1 RQ1: Impact of the Number of Edits

In this subsection, we provide experimental results details of the Mistral-7B model in Figure 11.

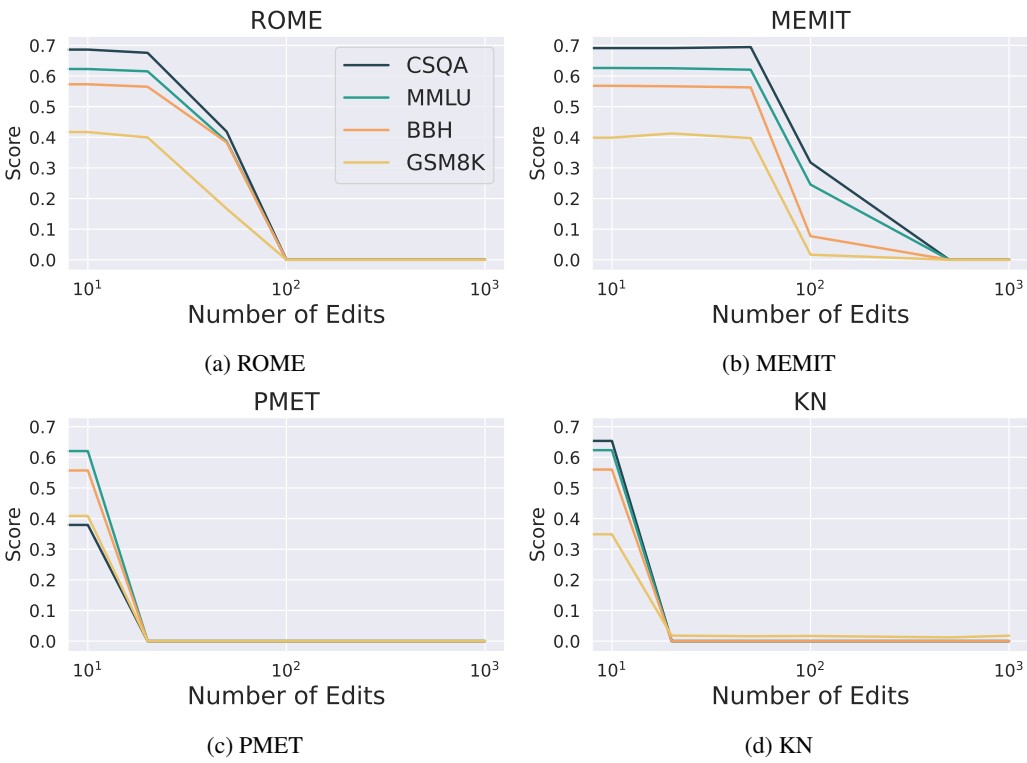

Figure 11: Performance trends of evaluating edited Mistral-7B based model across different benchmarks using 4 editing methods. Results reveal that all of the methods can't preserve the model's abilities across all tasks. While KN drastically drops even less than ten edits.

## B.2 RQ2: Does LLM with Instruction Tuning Show Better Performance after Editing?

In this subsection, we provide experimental results details of the Mistral-Instruct-7B model in Figure 12.

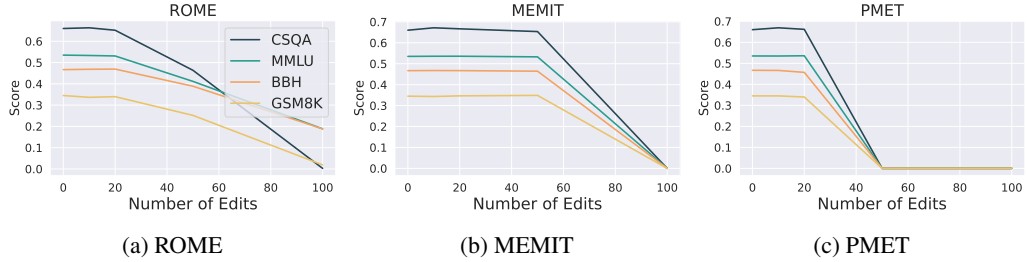

Figure 12: Performance trends of evaluating edited Mistral-Instruction-7B based model across different benchmarks using 4 editing methods. Results reveal that all of the methods can't preserve the model's abilities across all tasks. While KN drastically drops even less than ten edits.

# C   Detailed Experimental Results

In this section, we would like to present detailed experimental results of each research question.

## C.1   RQ1: Impact of the Number of Edits

In this subsection, we provide experimental results details of editing base models. The results of editing Llama2-7B and Mistral-7B can be found in Table 2, and the results for GPT2-XL can be found in Table 1.

| Method | # Edits | GPT2-XL | | | |
|---|---|---|---|---|---|
| | | MMLU | GSM8K | BBH | CSQA |
| w/o Edit | 0 | 0.2098 | 0.0144 | 0.0382 | 0.1941 |
| PMET | 10 | 0.2104 | 0.0159 | 0.0377 | 0.1941 |
| | 20 | 0.1081 | 0.0144 | 0.0117 | 0.2048 |
| | 50 | 0 | 0 | 0 | 0 |
| | 100 | 0 | 0 | 0 | 0 |
| | 500 | 0 | 0 | 0 | 0 |
| | 1000 | 0 | 0 | 0 | 0 |
| MEND | 10 | 0.2096 | 0.0144 | 0.0377 | 0.1949 |
| | 30 | 0.2094 | 0.0152 | 0.0388 | 0.1941 |
| | 100 | 0.2098 | 0.0144 | 0.0380 | 0.1957 |
| | 500 | 0.2100 | 0.0144 | 0.0382 | 0.1941 |
| | 1000 | 0.2099 | 0.0144 | 0.0381 | 0.1933 |
| KN | 500 | 0 | 0 | 0 | 0 |
| | 1000 | 0 | 0 | 0 | 0 |
| MEMIT | 500 | 0.2112 | 0.0159 | 0.0363 | 0.1957 |
| | 1000 | 0.2097 | 0.0152 | 0.0193 | 0.199 |

Table 1: Evaluation results of GPT2-XL. experiments are conducted on a sever with 8 RTX 4090 GPUs.

## C.2   RQ2: Does LLM with Instruction Tuning Show Better Performance after Editing?

In this subsection, we provide experimental results and details of editing chat models. The results of editing Llama2-chat-7B and Mistral-Instruct-7B can be found in Table 3.

## C.3   RQ3: Do the General Abilities of the Edited Model Differ on Model Scales?

In this subsection, we provide experimental results and details of exploring the impact of model scale on the general abilities of edited language models. The results of editing Pythia model families can be found in Table 4.

## C.4   RQ4: Does Editing Result in Variations in Different Aspects of a Model's Capabilities?

Nothing new here.

## C.5   RQ5: The Safety Cost of Editing Language Models

In this subsection, we provide experimental results and details of exploring the safety cost of editing language models. The results of editing Llama-7B, Llama-chat-7B, Mistral-7B, and Mistral-instruct-7B models can be found in Table 5.

| Method | # Edits | Llama2-7B | | | | Mistral-7B | | | |
|---|---|---|---|---|---|---|---|---|---|
| | | MMLU | GSM8K | BBH | CSQA | MMLU | GSM8K | BBH | CSQA |
| w/o Edit | 0 | 0.4587 | 0.1440 | 0.4000 | 0.5921 | 0.6233 | 0.3995 | 0.5670 | 0.6953 |
| ROME | 1 | 0.4581 | 0.1531 | 0.3991 | 0.5930 | 0.6246 | 0.4026 | 0.5710 | 0.6945 |
| | 5 | 0.4590 | 0.1418 | 0.3986 | 0.5856 | 0.6247 | 0.4086 | 0.5704 | 0.7027 |
| | 10 | 0.4572 | 0.1448 | 0.3970 | 0.5864 | 0.6229 | 0.4170 | 0.5730 | 0.6863 |
| | 20 | 0.4415 | 0.1486 | 0.3815 | 0.5627 | 0.6153 | 0.3995 | 0.5650 | 0.6757 |
| | 50 | 0.2701 | 0.0417 | 0.2835 | 0.2031 | 0.3864 | 0.1668 | 0.3835 | 0.4185 |
| | 100 | 0.0007 | 0.0159 | 0 | 0 | 0 | 0 | 0 | 0 |
| MEMIT | 1 | 0.4586 | 0.1456 | 0.4013 | 0.5921 | 0.6256 | 0.3965 | 0.5693 | 0.6896 |
| | 5 | 0.4569 | 0.1463 | 0.3975 | 0.5921 | 0.6256 | 0.3965 | 0.5679 | 0.6937 |
| | 10 | 0.4573 | 0.1501 | 0.3964 | 0.5946 | 0.6262 | 0.3988 | 0.5681 | 0.6912 |
| | 20 | 0.4578 | 0.1440 | 0.3944 | 0.5995 | 0.6254 | 0.4124 | 0.5665 | 0.6912 |
| | 50 | 0.4570 | 0.1509 | 0.3934 | 0.5913 | 0.6206 | 0.3973 | 0.5629 | 0.6945 |
| | 100 | 0.4428 | 0.0986 | 0.3767 | 0.5373 | 0.2456 | 0.0167 | 0.0771 | 0.3178 |
| | 500 | 0 | 0 | 0 | 0 | 0 | 0 | 0 | 0 |
| | 1000 | 0 | 0 | 0 | 0 | 0 | 0 | 0 | 0 |
| PMET | 1 | 0.4583 | 0.1456 | 0.3985 | 0.5913 | 0.6259 | 0.3958 | 0.5705 | 0.6904 |
| | 5 | 0.4586 | 0.1456 | 0.3998 | 0.5897 | 0.6256 | 0.4011 | 0.5690 | 0.6937 |
| | 10 | 0.4593 | 0.1494 | 0.4017 | 0.5930 | 0.6204 | 0.4086 | 0.5571 | 0.3792 |
| | 20 | 0.4588 | 0.1456 | 0.4013 | 0.5880 | 0 | 0 | 0 | 0 |
| | 50 | 0.4588 | 0.1448 | 0.4018 | 0.5880 | 0 | 0 | 0 | 0 |
| | 100 | 0.4593 | 0.1410 | 0.3962 | 0.5930 | 0 | 0 | 0 | 0 |
| | 500 | 0.4563 | 0.1380 | 0.3924 | 0.5856 | 0 | 0 | 0 | 0 |
| | 1000 | 0.4572 | 0.1319 | 0.3921 | 0.5823 | 0 | 0 | 0 | 0 |
| GRACE | 10 | 0.4272 | 0.1048 | 0.3872 | 0.5264 | - | - | - | - |
| | 20 | 0.3815 | 0.8860 | 0.3315 | 0.5027 | - | - | - | - |
| | 50 | 0.2401 | 0.0417 | 0.2635 | 0.1331 | - | - | - | - |
| | 100 | 0 | 0 | 0 | 0 | - | - | - | - |
| | 500 | 0 | 0 | 0 | 0 | - | - | - | - |
| | 1000 | 0 | 0 | 0 | 0 | - | - | - | - |
| MEND | 1 | 0.4573 | 0.1539 | 0.3994 | 0.5823 | - | - | - | - |
| | 5 | 0.4573 | 0.1494 | 0.3990 | 0.5905 | - | - | - | - |
| | 10 | 0.4573 | 0.1478 | 0.3945 | 0.5971 | - | - | - | - |
| | 20 | 0.4577 | 0.1486 | 0.3972 | 0.5905 | - | - | - | - |
| | 50 | 0.4586 | 0.1509 | 0.3992 | 0.5905 | - | - | - | - |
| | 100 | 0.4581 | 0.1531 | 0.3941 | 0.5856 | - | - | - | - |
| | 500 | 0.4577 | 0.1486 | 0.3992 | 0.5774 | - | - | - | - |
| | 1000 | 0.4571 | 0.1600 | 0.3978 | 0.5864 | - | - | - | - |
| KN | 10 | 0 | 0.0015 | 0 | 0 | 0.6234 | 0.3487 | 0.5601 | 0.6536 |
| | 50 | 0 | 0 | 0 | 0 | 0 | 0.0159 | 0 | 0 |
| | 100 | 0 | 0 | 0 | 0 | 0 | 0.0167 | 0 | 0 |
| | 500 | 0 | 0 | 0 | 0 | 0 | 0.0121 | 0 | 0 |
| | 1000 | 0 | 0 | 0 | 0 | 0 | 0.0175 | 0 | 0 |

Table 2: Results on evaluating the impact of different editing methods and numbers of edits on edited language models (base model). All editing is conducted on COUNTERFACT dataset with a fixed seed for a fair comparison. For all 4 tasks in this table, the higher score indicates a better performance. MEND and GRACE are not available for Mistral-7B.

| Method | # Edits | Llama2-7B-Chat | | | | Mistral-7B-Instruct | | | |
|---|---|---|---|---|---|---|---|---|---|
| | | MMLU | GSM8K | BBH | CSQA | MMLU | GSM8K | BBH | CSQA |
| w/o Edit | 0 | 0.4516 | 0.2032 | 0.3997 | 0.6134 | 0.5350 | 0.3450 | 0.4668 | 0.6601 |
| ROME | 1 | 0.4576 | 0.1531 | 0.3985 | 0.5938 | 0.5364 | 0.3442 | 0.4667 | 0.6699 |
| | 5 | 0.4587 | 0.1425 | 0.3976 | 0.5839 | 0.5354 | 0.3442 | 0.4648 | 0.6618 |
| | 10 | 0.4578 | 0.1471 | 0.3974 | 0.5864 | 0.5333 | 0.3366 | 0.4684 | 0.6634 |
| | 20 | 0.4416 | 0.1471 | 0.3828 | 0.5602 | 0.5310 | 0.3397 | 0.4693 | 0.6519 |
| | 50 | 0.2700 | 0.0409 | 0.2838 | 0.2048 | 0.4115 | 0.2517 | 0.3888 | 0.4636 |
| | 100 | 0.0007 | 0.0152 | 0 | 0 | 0.1884 | 0.0190 | 0.1884 | 0.0026 |
| MEMIT | 1 | 0.4715 | 0.2085 | 0.4106 | 0.6143 | 0.5356 | 0.3450 | 0.4664 | 0.6683 |
| | 5 | 0.4717 | 0.1895 | 0.4114 | 0.6233 | 0.5345 | 0.3419 | 0.4656 | 0.6675 |
| | 10 | 0.4704 | 0.2047 | 0.4132 | 0.6151 | 0.5357 | 0.3434 | 0.4674 | 0.6716 |
| | 20 | 0.4698 | 0.1956 | 0.4087 | 0.6405 | 0.5358 | 0.3465 | 0.4670 | 0.6667 |
| | 50 | 0.4682 | 0.2039 | 0.4017 | 0.6405 | 0.5328 | 0.3487 | 0.4643 | 0.6536 |
| | 100 | 0.4485 | 0.1850 | 0.3959 | 0.6044 | 0 | 0 | 0 | 0 |
| PMET | 1 | 0.4583 | 0.1471 | 0.3988 | 0.5930 | 0.5357 | 0.3465 | 0.6658 | 0.4663 |
| | 5 | 0.4586 | 0.1448 | 0.4001 | 0.5897 | 0.5356 | 0.3457 | 0.6691 | 0.4669 |
| | 10 | 0.4593 | 0.1471 | 0.4017 | 0.5930 | 0.5348 | 0.3450 | 0.6691 | 0.4662 |
| | 20 | 0.4588 | 0.1456 | 0.4010 | 0.5872 | 0.5360 | 0.3397 | 0.6618 | 0.4570 |
| | 50 | 0.4584 | 0.1448 | 0.4019 | 0.5905 | 0 | 0 | 0 | 0 |
| | 100 | 0.4590 | 0.1448 | 0.3960 | 0.5930 | 0 | 0 | 0 | 0 |
| MEND | 10 | 0.4731 | 0.2100 | 0.4097 | 0.6216 | - | - | - | - |
| | 20 | 0.4729 | 0.2024 | 0.4057 | 0.6102 | - | - | - | - |
| | 50 | 0.4728 | 0.2024 | 0.4101 | 0.6183 | - | - | - | - |
| | 100 | 0.4731 | 0.2009 | 0.4093 | 0.6183 | - | - | - | - |
| | 200 | 0.4738 | 0.2100 | 0.4030 | 0.6249 | - | - | - | - |
| | 500 | 0.4732 | 0.2168 | 0.4089 | 0.6192 | - | - | - | - |
| | 1000 | 0.4728 | 0.2138 | 0.4118 | 0.6224 | - | - | - | - |
| KN | 10 | 0 | 0 | 0 | 0 | 0 | 0 | 0 | 0 |
| | 20 | 0 | 0 | 0 | 0 | 0 | 0 | 0 | 0 |
| | 50 | 0 | 0 | 0 | 0 | 0 | 0 | 0 | 0 |

Table 3: Results on evaluating the impact of editing methods and numbers of edits on edited language models (Instruction tuned model). All editing is conducted on COUNTERFACT dataset with a fixed seed for a fair comparison. For all 4 tasks in this table, the higher score indicates a better performance.

| Model | Method | # Edits | MMLU↑ | GSM8K↑ | BBH↑ | CSQA↑ |
|---|---|---|---|---|---|---|
| Pythia-160M | w/o Edit | 0 | 0.2435 | 0.0174 | 0.0742 | 0.1884 |
| | ROME | 10 | 0 | 0 | 0 | 0 |
| | | 50 | 0 | 0 | 0 | 0 |
| | | 100 | 0 | 0 | 0 | 0 |
| | MEMIT | 10 | 0.2460 | 0.0212 | 0.0785 | 0.2056 |
| | | 50 | 0.2447 | 0.0227 | 0.0755 | 0.1982 |
| | | 100 | 0.2468 | 0.0235 | 0.0743 | 0.1990 |
| Pythia-410M | w/o Edit | 0 | 0.2614 | 0.0144 | 0.2497 | 0.2064 |
| | ROME | 10 | 0 | 0 | 0 | 0 |
| | | 50 | 0 | 0 | 0 | 0 |
| | | 100 | 0 | 0 | 0 | 0 |
| | MEMIT | 10 | 0.2628 | 0.0182 | 0.2476 | 0.2015 |
| | | 50 | 0.2629 | 0.0144 | 0.2482 | 0.2080 |
| | | 100 | 0.2627 | 0.0190 | 0.2490 | 0.2048 |
| Pythia-1B | w/o Edit | 0 | 0.2552 | 0.0273 | 0.2535 | 0.1892 |
| | ROME | 10 | 0.2547 | 0.0083 | 0.0052 | 0.2039 |
| | | 50 | 0.0017 | 0 | 0 | 0 |
| | | 100 | 0 | 0 | 0 | 0 |
| | MEMIT | 10 | 0.2562 | 0.0265 | 0.2545 | 0.1908 |
| | | 50 | 0.2539 | 0.0265 | 0.2544 | 0.2015 |
| | | 100 | 0.2547 | 0.0258 | 0.2532 | 0.2064 |
| Pythia-2.8B | w/o Edit | 0 | 0.2800 | 0.0364 | 0.2870 | 0.2146 |
| | ROME | 10 | 0.2272 | 0.0008 | 0.0004 | 0.1990 |
| | | 50 | 0.0001 | 0.0191 | 0 | 0 |
| | | 100 | 0 | 0 | 0 | 0 |
| | MEMIT | 10 | 0.2547 | 0.0303 | 0.2774 | 0.2154 |
| | | 50 | 0.2554 | 0.0349 | 0.2758 | 0.2269 |
| | | 100 | 0.2559 | 0.0318 | 0.2749 | 0.2179 |
| Pythia-6.9B | w/o Edit | 0 | 0.2565 | 0.0318 | 0.2762 | 0.2260 |
| | ROME | 10 | 0.0189 | 0 | 0 | 0 |
| | | 50 | 0 | 0 | 0 | 0 |
| | | 100 | 0 | 0 | 0 | 0 |
| | MEMIT | 10 | 0.2547 | 0.0303 | 0.2774 | 0.2154 |
| | | 50 | 0.2554 | 0.0349 | 0.2758 | 0.2269 |
| | | 100 | 0.2559 | 0.0318 | 0.2749 | 0.2179 |
| Pythia-12B | w/o Edit | 0 | 0.2621 | 0.0485 | 0.2868 | 0.2375 |
| | ROME | 10 | 0.0263 | 0.0380 | 0 | 0 |
| | | 50 | 0 | 0.0380 | 0 | 0 |
| | | 100 | 0 | 0.0380 | 0 | 0 |
| | MEMIT | 10 | 0.2615 | 0.0462 | 0.2878 | 0.2408 |
| | | 50 | 0.2633 | 0.0531 | 0.2916 | 0.2514 |
| | | 100 | 0.2587 | 0.0523 | 0.2925 | 0.2465 |

Table 4: Quantitative results of exploring the impact of model scale on post-edit large language models with Pythia language model families. We perform editing on COUNTERFACT datasets with different edits, then conduct evaluations on 4 benchmarks.

| Method | # Edits | Llama2-7B | | Llama2-7B-chat | | Mixtral-7B | | Mixtral-7B-Instruct | |
|---|---|---|---|---|---|---|---|---|---|
| | | TruthfulQA | Toxigen | TruthfulQA | Toxigen | TruthfulQA | Toxigen | TruthfulQA | Toxigen |
| w/o Edits | 0 | 0.2521 | 0.4284 | 0.3023 | 0.5177 | 0.2815 | 0.4247 | 0.3917 | 0.4896 |
| | 1 | 0.2521 | 0.4296 | 0.2921 | 0.5196 | 0.2815 | 0.4247 | 0.3941 | 0.4810 |
| | 5 | 0.2497 | 0.4272 | 0.2997 | 0.5072 | 0.2815 | 0.4247 | 0.3929 | 0.4896 |
| ROME | 10 | 0.2485 | 0.4296 | 0.2962 | 0.5080 | 0.2742 | 0.4235 | 0.3892 | 0.4737 |
| | 20 | 0.2411 | 0.4284 | 0.2913 | 0.4871 | 0.2742 | 0.4247 | 0.3868 | 0.4737 |
| | 50 | 0.2411 | 0.4101 | 0.2497 | 0.4957 | 0.2350 | 0.4247 | 0.2644 | 0.4504 |
| | 100 | 0.2729 | 0.4982 | 0.2974 | 0.5141 | 0.2509 | 0.5667 | 0.2827 | 0.5251 |
| | 1 | 0.2509 | 0.4284 | 0.2999 | 0.5116 | 0.2815 | 0.4272 | 0.3905 | 0.4859 |
| | 5 | 0.2497 | 0.4272 | 0.2950 | 0.5116 | 0.2803 | 0.4272 | 0.3929 | 0.4908 |
| MEMIT | 10 | 0.2497 | 0.4284 | 0.2925 | 0.5153 | 0.2815 | 0.4259 | 0.3929 | 0.4847 |
| | 20 | 0.2460 | 0.4308 | 0.2999 | 0.5018 | 0.2791 | 0.4259 | 0.3917 | 0.4908 |
| | 50 | 0.2399 | 0.4308 | 0.2815 | 0.5153 | 0.2668 | 0.4308 | 0.3807 | 0.4774 |
| | 100 | 0.1922 | 0.4321 | 0.2472 | 0.4896 | 0.2375 | 0.4627 | 0.2350 | 0.5838 |
| | 1 | 0.2521 | 0.4296 | 0.2974 | 0.5163 | 0.2815 | 0.4247 | 0.3917 | 0.4823 |
| | 5 | 0.2497 | 0.4272 | 0.2988 | 0.5175 | 0.2815 | 0.4247 | 0.3917 | 0.4835 |
| | 10 | 0.2485 | 0.4296 | 0.2964 | 0.5190 | 0.2840 | 0.4235 | 0.3929 | 0.4847 |
| PMET | 20 | 0.2411 | 0.4284 | 0.2974 | 0.5141 | 0.2740 | 0.4247 | 0.3905 | 0.4908 |
| | 50 | 0.2411 | 0.4100 | 0.2962 | 0.5129 | 0.2350 | 0.4247 | 0.2375 | 0.4333 |
| | 100 | 0.2729 | 0.4982 | 0.2962 | 0.5165 | 0.2509 | 0.5667 | 0.2350 | 0.4333 |
| | 500 | 0.2350 | 0.4259 | 0.2362 | 0.5667 | - | - | - | - |
| | 1000 | 0.2362 | 0.4308 | 0.2350 | 0.5667 | - | - | - | - |
| | 10 | 0.2472 | 0.4308 | 0.2974 | 0.5141 | - | - | - | - |
| | 20 | 0.2546 | 0.4296 | 0.2999 | 0.5104 | - | - | - | - |
| MEND | 50 | 0.2521 | 0.4296 | 0.2938 | 0.5153 | - | - | - | - |
| | 100 | 0.2521 | 0.4296 | 0.3035 | 0.5153 | - | - | - | - |
| | 500 | 0.2521 | 0.4308 | 0.3035 | 0.5080 | - | - | - | - |
| | 1000 | 0.2485 | 0.4308 | 0.2950 | 0.5055 | - | - | - | - |
| | 10 | 0.2350 | 0.4333 | 0.2277 | 0.4333 | 0.2889 | 0.4308 | - | - |
| | 50 | 0.2399 | 0.5667 | 0.2399 | 0.4590 | 0.2558 | 0.5667 | - | - |
| KN | 100 | 0.2350 | 0.5667 | 0.2399 | 0.4590 | 0.2583 | 0.5667 | - | - |
| | 500 | 0.2362 | 0.4333 | 0.2392 | 0.4590 | 0.2583 | 0.5667 | - | - |
| | 1000 | 0.2313 | 0.4333 | 0.2399 | 0.4590 | 0.2583 | 0.5667 | - | - |

Table 5: Quantitative analysis on language model safety after undergoing edits. We perform editing on COUNTERFACR dataset with fixed seed for fair comparison.

# D  Evaluation Benchmark

## D.1  Dataset Details

For our evaluation benchmark, we employ seven datasets into two distinct evaluation methodologies: generation-based and sequence-based [76]. The generation-based method utilizes vLLM [47] inference framework and following the procedures outlined in Chain-of-Thought Hub [77] and Active-Prompt [78]. For sequence-based evaluations, we use the Language Model Evaluation Harness framework [79]. Detailed statistics for each benchmark dataset are provided in Table 6.

| Dataset | Task Type | # Few-shot | # Test | Metric | Evaluation Method |
|---|---|---|---|---|---|
| MMLU [28] | World Knowledge | 5 | 14,079 | Accuracy | Generation-Based |
| BBH [29] | World Knowledge | 3 | 6,511 | Accuracy | Generation-Based |
| GSM8K [40] | Arithmetic | 8 | 1,319 | Exact match | Generation-Based |
| CSQA* [41] | Commonsense | 7 | 1,221 | Accuracy | Generation-Based |
| TriviaQA [42] | Reading Comprehension | 0 | 17,900 | Exact match | Generation-Based |
| TruthfulQA [43] | Truthful | 0 | 817 | Accuracy | Sequence-Based |
| ToxiGen [44] | Hate Speech | 0 | 940 | Accuracy | Sequence-Based |

Table 6: The statistics of the datasets used in this paper. # Ex. are the number of few-shot chain-of-thought exemplars used to prompt each task in evaluation. # Test denote the number of training data and test data, respectively. *: CSQA do not have publicly available test set labels, so we simply follow the setting by [80; 78] to evaluate the performance of the development set.

## D.2  Evaluation Efficiency

Certain model editing methods, such as GRACE [38] and SERAC [18], modify the model architecture, which can result in incompatibilities with serving frameworks like vLLM. To provide a clear understanding of the efficiency impact, Table 7 compares the time costs of running benchmarks with and without vLLM. The comparison highlights the significant reduction in time costs when using vLLM, demonstrating its efficiency. Specifically, for the Llama2-7B model, the MMLU benchmark shows a reduction from 840 minutes to 103 minutes, the GSM8K benchmark from 7 minutes to 5 minutes, and the CSQA benchmark from 42 minutes to 26 minutes.

| Method | With vLLM | | | Without vLLM | | |
|---|---|---|---|---|---|---|
| | MMLU | GSM8K | CSQA | MMLU | GSM8K | CSQA |
| Llama2-7B | 103 | 5 | 26 | 840 | 7 | 42 |

Table 7: Comparison of time costs for different benchmarks with and without vLLM using the Llama2-7B model. The unit is minutes. The table demonstrates that using vLLM significantly reduces the time costs across all benchmarks.

# E  Editing Efficiency

In this section, we provide the editing experiment running time statics in Table 8.

| Method | Llama2-7B | | | GPT2-XL | | |
|--------|-----|-----|------|-----|-----|------|
|        | 10  | 50  | 100  | 10  | 50  | 100  |
| ROME   | 2m1s   | 9m53s   | 16m31s | 59s   | 4m4s   | 8m11s  |
| MEMIT  | 4m30s  | 20m29s  | 40m14s | 2m10s | 8m24s  | 17m23s |
| GRACE  | 10s    | 1m3s    | 2m1s   | 5s    | 31s    | 1m2s   |
| MEND   | 24s    | 1m34s   | 2m17s  | 11s   | 52s    | 1m24s  |
| SERAC  | 20s    | 1m7s    | 1m24s  | 14s   | 1m12s  | 2m15s  |

Table 8: Execution time of different editing methods on Llama2-7B and GPT2-XL model with different edits. All of the experiments are conducted on a single GPU.

# F  Implementation and Reproduction Details

In this section, we would like to provide details for reproducing our experimental results.

## F.1  Code Base

Here, we list the code base used in our paper.

- For all of the models, we use huggingface transformers as default `https://github.com/huggingface/transformers`.
- For editing language models, we use EasyEdit framework `https://github.com/zjunlp/EasyEdit`.
- For model evaluation, we use the code and data from chain-of-thought hub `https://github.com/FranxYao/chain-of-thought-hub` and Language Model Evaluation Harness `https://github.com/EleutherAI/lm-evaluation-harness`.
- For accelerating model evaluation on GPU, we use the vLLM framework `https://github.com/vllm-project/vllm`.

Our code is available at `https://github.com/lqinfdim/EditingEvaluation`.

## F.2  Models

Here, we list all of the model checkpoints used in our experiments.

- Llama2-7B `https://hf-mirror.com/meta-llama/Llama-2-7b-hf`
- Llama2-chat-7B `https://hf-mirror.com/meta-llama/Llama-2-7b-chat-hf`
- Mistral-7B `https://hf-mirror.com/mistralai/Mistral-7B-v0.1`
- Mistral-Instruct-7B `https://hf-mirror.com/mistralai/Mistral-7B-Instruct-v0.1`
- GPT2-XL `https://hf-mirror.com/openai-community/gpt2-xl`
- Pythia-160M `https://hf-mirror.com/EleutherAI/pythia-160m`
- Pythia-410M `https://hf-mirror.com/EleutherAI/pythia-410m`
- Pythia-1.4B `https://hf-mirror.com/EleutherAI/pythia-1.4b`
- Pythia-2.8B `https://hf-mirror.com/EleutherAI/pythia-2.8b`
- Pythia-6.9B `https://hf-mirror.com/EleutherAI/pythia-6.9b`
- Pythia-12B `https://hf-mirror.com/EleutherAI/pythia-12b`

### F.3 Hyperparameter Settings

In editing experiments, our hyperparameters are in accord with the EasyEdit framework at `https://github.com/zjunlp/EasyEdit/tree/main/hparams`.

For model evaluation, please refer to Appendix D.

### F.4 Running Devices

All of our experiments are running on two devices: a server with 8 RTX A6000 GPUs with 48GB VRAM, and another server equipped with 8 RTX 4090 GPUs with 24GB VRAM.

