# OpenReview forum: "Should We Really Edit Language Models? On the Evaluation of Edited Language Models"
_NeurIPS.cc/2024/Conference — NeurIPS 2024 poster_

### Official Review · Reviewer_FFCf · 2024-07-12

**Soundness:** 4
**Presentation:** 4
**Contribution:** 3
**Rating:** 5
**Confidence:** 2

**Summary:**

The paper explores the general abilities of post-edited language models. Concretely, the paper performs a comprehensive empirical evaluation no various model editing methods and language models. The paper summarizes key findings based on the number of edits, the scale of the language model, the type of tuning, and the safety. The paper shows various numerical metrics to support these findings. The supplementary provides detailed results for experiments and the benchmark construction.

**Strengths:**

1. The paper is well-written, precise, and easy to follow. All findings are summarized and listed.

2. The experiments are comprehensive and detailed. The related dataset, implementations, and evaluation metrics are provided in detail and well-organized.

3. The evaluation includes several latest language models and editing methods which are useful for later research.

**Weaknesses:**

1. Though the experiments are comprehensive and organized, the findings are more about empirical observations rather than systematic analysis. The paper does not distill or form a theory or systematic justification based on these findings. Thus, it is hard to judge academic contributions, especially in terms of the criterion of Neurips.

2. Though the experiments are comprehensive and organized, the findings (line155~159) are more about empirical observations that do not form systematic logic.

3. The paper does not propose any new algorithms to improve the sequential model editing.

**Questions:**

1. Is there any insight to improve the sequential model editing based on these findings?

2. In the experiments of sequential editing, the relation of these edits is not presented. For example, if there are 20 edits, does the correlation of edits in different order influence the conclusion and results?

**Limitations:**

The author does not provide any limitations for the evaluation method.

---

> ### Author Rebuttal · Authors · 2024-08-05
>
> Thank you for your time devoted to reviewing this paper and your constructive suggestions. Here are our detailed replies to your questions.
>
> > Q1: Though the experiments are comprehensive and organized, the findings are more about empirical observations rather than systematic analysis. The paper does not distill or form a theory or systematic justification based on these findings. Thus, it is hard to judge academic contributions, especially in terms of the criterion of Neurips.
>
> **Reply:** Thank you for your valuable questions. I want to address your concerns in two parts:
>
> **(1) Purpose and Motivation:**
>
> The primary objective of this paper is **not** to introduce a novel editing method. Instead, we aim to provide a detailed evaluation of existing knowledge editing methods' impact on LLM performance under the sequential editing setting. By investigating the potential factors influencing the performance of edited LLMs, we hope to offer insights and guidance for future research in this area.
>
> **(2) A more systematic and general view of our findings:"Elasticity and Plasticity Trade-off"**
>
> We have re-organized the findings of our paper and made the following analysis and observation , which we term the Elasticity and Plasticity Trade-off (EPT):
>
> First, we define two crucial concepts in sequential knowledge editing: plasticity, referring to the model's ability to retain newly acquired knowledge through editing, and elasticity, denoting the model's ability to retain its original knowledge after editing.
>
> We observe that during large-scale sequential editing, simultaneously maintaining high plasticity (retention of updated knowledge) and high elasticity (retention of inherent knowledge) is challenging. These two objectives are inherently in conflict. We refer to this phenomenon as the Elasticity and Plasticity Trade-off, drawing parallels to the Stability-Plasticity Dilemma observed in catastrophic forgetting during neural network fine-tuning. Existing work primarily focuses on the detrimental effect of large-scale sequential editing on elasticity, neglecting the interplay between elasticity and plasticity. Our work highlights this crucial trade-off, emphasizing the inherent conflict between these two objectives.
>
> Experimental results with different editing methods on Llama2-7B (see Table 1) demonstrate that when applying editing, maintaining a balance between elasticity and plasticity requires limiting the number of edits within a reasonable range. Exceeding it can lead to model collapse, characterized by the destruction of the model's intrinsic knowledge structure.
>
>  Results  in Table 1 also show that some methods can maintain the EPT over thousands of edits. However,  this capability is not unlimited. With a sufficiently large number of edits, model collapse still occurs. We introduce the concept of  **editing endurance** to quantify the maximum number of edits a model can undergo while preserving the EPT.
>
> | # Edits | MMLU  | BBH   | TrivialQA | Efficacy | Generalization | Locality |
> | :------ | :---- | :---- | :-------- | :------- | :------------- | :------- |
> | 0       | 0.459 | 0.400 | 0.525     | 17.4     | 19.2           | 86.1     |
> | 10      | 0.459 | 0.401 | 0.523     | 100      | 95.6           | 81.3     |
> | 100     | 0.459 | 0.396 | 0.521     | 100      | 95.5           | 81.6     |
> | 500     | 0.456 | 0.392 | 0.499     | 99.4     | 95.4           | 80.7     |
> | 1000    | 0.457 | 0.392 | 0.490     | 99.1     | 95.4           | 80.2     |
>
> **Table 1: Edit result for PMET**
>
> To provide an intuitive understanding, we examine the weight changes with different edits in edited layer (see Table 2). This analysis aims to demonstrate that maintaining EPT necessitates keeping weight changes within a reasonable range like PMET (neither too big nor too small).
>
> | Method | 0       | 10      | 100     | 1000     |
> | :----- | :------ | :------ | :------ | :------- |
> | ROME   | 117.053 | 118.265 | 269.497 | overflow |
> | MEMIT  | 116.579 | 116.911 | 121.681 | 9291.127 |
> | PMET   | 116.579 | 116.558 | 116.667 | 117.749  |
> | MEND   | 126.209 | 126.198 | 126.198 | 126.199  |
>
> **Table 2: L2 Norm of edited layer for different edits on Llama2-7B model.**
>
> **Please refer to general response part for more detailed explanation or experiments of EPT.**
>
> In summary, we believe that the research questions we address are of significant importance to editing. This work will enhance the understanding of knowledge editing for the research community.
>
> > Q2: Though the experiments are comprehensive and organized, ... form systematic logic.
>
> **Reply:**  We would like to clarify that our intuition is to (lines 155-159) pose research questions aimed at exploring common factors that might influence the performance of edited models.
>
> To enhance the systematic nature of our work, we unify our findings under the framework of the Elasticity and Plasticity Trade-off (as elaborated in the response to Q1). Consequently, the five questions posed in lines 155-159 can be considered an analysis and understanding of the **editing endurance** of the model.
>
>
> > Q3: The paper does not propose any new algorithms to improve the sequential model editing.
>
> **Reply:**  We would like to clarify that the primary objective of this paper is to provide a comprehensive evaluation of the impact of existing knowledge editing methods on LLM performance within the context of sequential editing. We aim to investigate potential factors influencing performance, offering valuable insights and guidance for future research in this area. Our focus is **not** on proposing specific new methods.
>
> Given the primary area of submission for this paper is **evaluation**, and considering the requirements of this area, we believe that our focus on evaluating and analyzing existing methods, rather than improving specific algorithms, is appropriate. We kindly request that you take this into careful consideration.

---

> ### Author Response · Authors · 2024-08-13
> **Kindly Reminder**
>
> Dear Reviewer FFCf:
>
> Thank you very much for your dedicated review. In the rebuttal period, we provide detailed responses to all your comments and questions point-by-point.
>
> A brief summary of our responses includes:
>
> + Q1: A more systematic and general view of our findings: "Elasticity and Plasticity Trade-off".
> + Q2: A boarder view of research questions.
> + Q3: Why are there no new algorithms in our paper.
>
> If there are any remaining issues or further clarifications needed, please let us know. We are more than happy to provide additional information or explanations to help resolve any concerns.
>
> Thank you for your time and valuable feedback.
>
> Best regards

---

> > ### Comment · Reviewer_FFCf · 2024-08-14
> > **Response to the rebuttal**
> >
> > Thanks for the clarification and additional results!
> >
> > The results have addressed my questions. I choose to raise my score from 4 to 5.

---

### Official Review · Reviewer_mS2B · 2024-07-13

**Soundness:** 3
**Presentation:** 3
**Contribution:** 2
**Rating:** 6
**Confidence:** 4

**Summary:**

The work evaluates the impact of various editing methods on LLMs. Specifically, it investigates how different editing techniques affect the general abilities of models, considering factors such as the number of edits, model scale, safety, and different aspects of model capabilities. Interesting findings and conclusions are suggested by the results.

**Strengths:**

+ The work effectively outlines the problem statement related to editing language models and provides a clear framework for evaluating the impact of different editing methods on model performance. Moreover, the study includes a thorough literature review on LLM knowledge editing.

+ The empirical studies conducted in the research are solid and rigorous. The study offers reliable findings that can inform future research and development in the field.

+ Safety is a great perspective to study the current challenges of LLM editing.

**Weaknesses:**

- The outcome of the empirical study is somehow limited and can be further extended. The results are pretty intuitive and do not surprise people much from the current understanding. The authors can provide further insights on the following aspects. 1) Among different editing methods, what are really being traded off when number of editing increases (e.g., does PMET trade editing performance to preserve more general capabilities? And why?). It’d be great to see some hypothesis/quantification to identify and push the boundaries of performance tradeoff. The outcome will be generalizable to future development of methods. 2) Also, given the results, the audience may be further interested in why the muting effect could happen and why instruction-tuned models are more robust. Performing deeper analysis and find associations on the effects would add more value to the results.

- The studied problem can be better motivated. If edit needs to be done at scale, e.g., more than 1k edit, is LLM editing still desired or a refreshed fine-tuning could already do the job better? Longer context and retrieval based generation for time sensitive knowledge is also usually consider as common solutions, which may weaken the need for sequential editing at scale.

- For methods such as PMET and MEND that are robust within 10^3 editing, could the authors further extend the scale of # edits to verify if the a similar drastic performance drop, as well as the muting effect happens?

- The term “intrinsic knowledge structure” appears multiple times to explain the potential reason. The claims can be more solid if a definition and detail discussion can be provided regarding the knowledge structure.

**Questions:**

Discussed in weaknesses.

---

> ### Author Rebuttal · Authors · 2024-08-05
>
> Thank you for your time devoted to reviewing this paper and your constructive suggestions. Here are our detailed replies to your questions.
>
> > Q1: The outcome of the empirical study is somehow .... would add more value to the results.
>
> **Reply:** Thanks for your suggestion. we have re-organized the findings of our paper and made the following analysis and observation , which we term the **Elasticity and Plasticity Trade-off (EPT)**:
>
> First, we define two crucial concepts in sequential knowledge editing: **plasticity**, referring to the model's ability to retain newly acquired knowledge through editing, and **elasticity**, denoting the model's ability to retain its original knowledge after editing.
>
> We observe that during large-scale sequential editing, simultaneously maintaining high plasticity and high elasticity  is challenging. These two objectives are inherently in conflict. We refer to this phenomenon as the **Elasticity and Plasticity Trade-off**, drawing parallels to the Stability-Plasticity Dilemma observed in catastrophic forgetting during neural network fine-tuning.
>
> When editing, maintaining a balance between elasticity and plasticity requires limiting the number of edits within a reasonable range. Exceeding this limit can lead to model collapse. We introduce the concept of editing endurance to quantify this limit. Please refer to **general response part** for detailed experiments and analysis.
>
> > Q2: The studied problem can be better motivated. If edit needs to be done at scale, e.g., more than 1k edit, is LLM editing still desired or a refreshed fine-tuning could already do the job better? Longer context and retrieval-based generation for time-sensitive knowledge is also usually considered as common solutions, which may weaken the need for sequential editing at scale.
>
> **Reply:** We would like to clarify that the primary objective of this paper is to provide a comprehensive evaluation of the impact of existing editing methods on LLM performance. We aim to investigate potential factors influencing performance, offering valuable insights and guidance for future research in this area.
>
> Indeed, each approach like FT, and RAG possesses unique advantages, necessitating careful selection based on the specific application scenario. Our work strives to equip researchers and engineers with a thorough understanding of knowledge editing's strengths, limitations, and performance implications, enabling them to choose the most appropriate approach for modifying model behavior.
>
> Here's a brief summary of the pros and cons of the related techniques you mentioned:
>
> **Fine-tuning:**
>
> *   **Pros:** Suitable for integrating a large amount of new knowledge into the model.
> *   **Cons:** Resource-intensive and prone to catastrophic forgetting.
>
> **Retrieval Augmented Generation (RAG):**
>
> *   **Pros:** Enables rapid knowledge updates, provides domain-specific knowledge.
> *   **Cons:** Can lead to hallucinations when retrieved knowledge conflicts with the model's internal knowledge. Retrieved text may not always be relevant to the desired topic.
>
> **Knowledge Editing :**
>
> *   **Pros:** Offers fine-grained and efficient modification of model knowledge.
> *   **Cons:** Existing methods only support a limited number of knowledge edits. Edited knowledge is difficult to utilize for reasoning tasks.
>
> **Long-Context :**
>
> *   **Pros:** Allows models to retrieve and cross-verify information within a long context, potentially correcting errors or inconsistencies and enabling more complex reasoning.
> *   **Cons:** Requires additional training, making it extremely resource-intensive.  Can also lead to hallucinations when prompt knowledge conflicts with the model's inherent knowledge.
>
> > Q3: For methods such as PMET and MEND that are robust within 10^3 editing, could the authors further extend the scale of # edits to verify if a similar drastic performance drop, as well as the muting effect happens?
>
> **Reply:** Thank you for your question. To further explore the limitations of the PMET and MEND methods, we extended the number of experiments to 3K on Llama2-7B model. The results are as follows:
>
> |Method| # Edits | MMLU | GSM8K | BBH | CSQA |
> |:-:|:------:|:--:|:--:|:---:|:----:|
> |w/o Edit|0   |  0.4587  |  0.1440  |  0.4000   |   0.5921   |
> | PMET | 1000  |   0.4572 |  0.1391  |  0.3921   |  0.5823    |
> |PMET | 2000 | 0 | 0 |0 | 0|
> | MEND | 1000  |   0.4571 |  0.1600  |  0.3978   |  0.5864    |
> |MEND | 2000 | 0.4581 | 0.1501 |0.4014 | 0.5905|
> |MEND | 3000 | 0.4574 | 0.1539 |0.3903 | 0.5667|
>
> Table 1: scaling edits to 3k
>
> As we can see from above table, PMET shows the muting effect. Although MEND still maintains relatively good performance, this does not indicate that it has good performance. Instead, it has high elasticity and low plasticity.
>
> > Q4: The term “intrinsic knowledge structure” ... structure.
>
> **Reply:** We would like to clarify that "Intrinsic knowledge structure" is an intuitive understanding of how knowledge is stored and organized within LLMs.
>
> Extensive existing work [1,2,3,4] suggests that parameter in LLMs implicitly encode knowledge. Manipulating these weights consequently affects the stored knowledge. A line of works[5,6,7]  precisely alter the behavior of model by change parameters. Such alteration can cause weight distribution shift.
>
> When weight distribution changes induced by such manipulations remain within a reasonable range, the model retains its original knowledge. However, if these weight changes become too drastic, the model's distribution collapses, making it challenging to extract new knowledge.
>
> ---
> ## Reference
>
> [1] Locating and Editing Factual Associations in GPT
>
> [2] Transformer Feed-Forward Layers Are Key-Value Memories
>
> [3] Knowledge Neurons in Pretrained Transformers
>
> [4] Kformer: Knowledge Injection in Transformer Feed-Forward Layers
>
> [5] Mass-Editing Memory in a Transformer
>
> [6] PMET: Precise Model Editing in a Transformer
>
> [7] Editing Common Sense in Transformers

---

> > ### Comment · Reviewer_2vUP · 2024-08-12
> > **Thank you for your rebuttal**
> >
> > Dear Authors,
> >
> > Thank you for your thorough rebuttal. I will update my review to raise my score.
> >
> > However, I have some lingering questions that you might consider addressing in your paper to provide more clarity and understanding: Is model editing necessary in the age of APIs? If so, who is going to perform model editing at all? In what situations should model editing be even considered aside from FT or RAGs?

---

> > > ### Author Response · Authors · 2024-08-13
> > > **Thanks  for response**
> > >
> > > Dear Reviewer mS2B:
> > >
> > > We are glad that our responses address some of your concerns.
> > >
> > > Regarding the question you raised about the use cases for model editing: We believe that in the era of large models and APIs, model editing has become increasingly necessary, especially when resources are limited or when there are few errors that don’t justify full fine-tuning but still need correction. Consider these typical scenarios:
> > >
> > > 1. If the entity deploying the model is different from the one training it, the deploying entity might lack sufficient computational resources to correct the model's errors through fine-tuning.
> > > 2. If the same organization is responsible for both training and deploying the model, but the model only has minor errors that don’t immediately warrant fine-tuning.
> > > 3. In situations that require quick responses, such as news updates or emergency events, model editing offers a rapid method of correction without the need for a complete fine-tuning process.
> > > 4. In highly sensitive or tightly regulated fields (e.g., finance, healthcare), model editing provides a precise and controlled way to correct specific errors without affecting other parts of the model.
> > >
> > > Regarding the use cases for RAG (Retrieval-Augmented Generation) and FT (Fine-Tuning): We believe that fine-tuning is suitable for correcting a large number of errors or injecting domain knowledge that the model has not previously learned. This approach can alter the model's overall behavior or style. RAG, on the other hand, is ideal for injecting domain knowledge into a model without fine-tuning, allowing for dynamic knowledge injection, especially for frequently changing information, without impacting the model's overall capabilities.
> > >
> > > We also welcome new suggestions/comments from you!
> > >
> > > Best regards,
> > >
> > > Author of paper 4258

---

### Official Review · Reviewer_2vUP · 2024-07-13

**Soundness:** 3
**Presentation:** 4
**Contribution:** 3
**Rating:** 6
**Confidence:** 4

**Summary:**

[Post rebuttal update] The authors have addressed my main concern about providing an overarching framework for their evaluation, using the Elasticity - Plasticity tradeoff. I will raise my score from 5 to 6.

This paper evaluates the influence of several model editing methods on the models' general capabilities. Their key dimensions of analysis include performance deterioration after several sequential single edits, the robustness of instruction-tuned models, the effect of LLM size and how editing mechanisms affect model safety. The paper stresses that current editing approaches are suitable only for minor updates and highlights the need for more practical and reliable editing techniques. They reveal that while a few edits have minimal impact, extensive edits can lead to significant degradation, including a phenomenon called muting effect, where models tend to produce empty outputs.

**Strengths:**

1. The study provides a thorough assessment of various model editing methods across different language models, offering valuable insights into their effects on general capabilities.
2. By highlighting the pitfalls of current editing methods, such as knowledge distortion and catastrophic forgetting, the paper addresses critical issues that need to be resolved for model editing approaches to work in practical deployments of LLMs. The research emphasizes the impact of editing on the overall performance of language models, including world knowledge, reading comprehension, reasoning, and safety, rather than just on specific tasks. This broader perspective is crucial for understanding the real-world implications of model editing.
3. The study’s exploration of how editing affects the safety of models is particularly valuable, as it addresses a key concern for deploying language models in sensitive applications.
4. The paper claims to be the first to comprehensively evaluate and analyze the general abilities of edited language models. If this claim is true, then it makes it a timely contribution.

**Weaknesses:**

1. The main weakness of the paper is in the exhaustive nature of the evaluation. While the paper evaluates multiple models, the findings may not generalize across all types of language models or specific applications, potentially limiting the broader applicability of the results. It is unfair to ask the authors to work with tons of LLMs but an application stratified view is missing.
2. The paper focuses more on empirical evaluation rather than providing deep technical insights into why certain methods cause performance degradation or safety issues. A more detailed technical analysis could enhance understanding and drive innovation. Note that by technical analysis, I do not mean theoretical analysis, but rather a deeper insight as to what issues are causing these failure modes. Are we not sanitizing the edits well? Are we overfitting? Is there an early stopping criterion that may work here? While pointing out flaws in existing methodologies is useful, since there are many such instances in the LLM world, well grounded research has to provide insights as to why the methodologies are flawed.
3. As an example of (2) above: The observation of the "muting effect," where models produce empty outputs after extensive edits, is concerning. However, the paper does not propose concrete insights as to why this issue might occur, leaving an important problem unaddressed.
4. There are also several different dimensions to consider for an analysis such as this: Scale of the model, prompting conditions or methodologies, whether models benefit from RAGs or not. Since the authors stress practical deployments, they should ensure their results are consistent with practical deployments as well.

**Questions:**

In addition to points above, I wonder what the author's thoughts are on what their best insights are as to why the model performance degrades? How would a practitioner deploying a complex LLM API go about figuring this out? consider cases where they have full models access or just API access.

**Limitations:**

In addition to addressing the weaknesses above, I encourage the authors to think of a more comprehensive evaluation framework to showcase their work. Evaluations of LLMs are themselves brittle -- hence any evaluation needs to have sufficient power to make a solid claim. In the current situation, the paper is indeed well written, but not statistically strong nor comprehensive.

---

> ### Author Rebuttal · Authors · 2024-08-05
>
> Thank you for your time devoted to reviewing this paper and your constructive suggestions. Here are our detailed replies to your questions.
>
> > Q1: The main weakness ... stratified view is missing.
>
> **Reply:** Thank you for your valuable feedback. We address your questions and concerns regarding the scope of our evaluation below:
>
> **(1) Can findings generalize to other types of language models?**
>
> Currently, the vast majority of methods in editing like ROME, MELLO, IKE, focus on decoder-only models (GPT, Llama).  The exploration of knowledge editing for other encoder-only (BERT) and encoder-decoder (T5) models  remains largely under-explored at present, making it infeasible to evaluate all kinds of model.
>
> **(2) Can findings generalize to specific application scenarios? **
>
> We want to clarify that editing is not tied to specific application scenarios. Similar to pre-training, SFT, and RLHF, it is a stage in the model whole life pipeline. Therefore, we did not evaluate on specific application scenarios. In fact, knowledge editing can be applied to LLMs across various applications, making our evaluation general and broadly applicable.
>
> To justify our conclusion in specific application scenarios, we chose QA task as example , conducting evaluation on TriviaQA dataset with Llama2-7B. The results are shown below, note the un-edited model gets 0.5252 on TriviaQA:
>
> | Method | 10 | 50 | 100 | 1000 |
> |--------|----|----|-----|------|
> | ROME   | 0.4887   |  0.0035  |  0   | 0     |
> | MEND   |  0.5288  |  0.5248  |  0.527   |   0.5282   |
> | PMET   |   0.5237 |  0.5199  |  0.5209   |  0.4904    |
> | MEMIT  |  0.5247  | 0.5189   |  0.2121   | 0     |
> | KN     |   0 |  0  | 0    |     0 |
>
> The trends  in the table are consistent with the conclusions  in our paper.
>
> > Q2: The paper focuses more on ... the methodologies are flawed.
>
> **Reply:** Thanks comment. We answer your question in two folds:
>
> (1) **Distinction between  editing and FT:**
>
> Knowledge editing differs from FT. Most editing methods in this paper are gradient-free, **without** something like early stop in FT. These methods typically bypass backpropagation and gradient descent, instead directly calculating and substituting specific parameters to achieve the desired behavior change.
>
> (2) **Deeper understanding and systematic analysis:**
>
> Please refer to Q3 for detailed explanation.
>
> > Q3: As an example of Q2 above ....
>
> **Reply:** Existing work suggests that parameter weights in LLMs represent specific knowledge. Manipulating these weights consequently affects the stored knowledge. Through extensive experimentation, we have made the following analysis and observation regarding editing, which we term the Elasticity and Plasticity Trade-off (EPT):
>
> First, we define two crucial concepts : **plasticity**, referring to the model's ability to retain newly acquired knowledge through editing, and **elasticity**, denoting the model's ability to retain its original knowledge after editing.
>
> We observe that during large-scale sequential editing, simultaneously maintaining plasticity  and high elasticity is challenging. These two objectives may inherently in conflict. We refer to this as the **Elasticity and Plasticity Trade-off**, drawing parallels to the Stability-Plasticity Dilemma  in catastrophic forgetting during fine-tuning.
>
> When editing, maintaining a balance between elasticity and plasticity requires limiting the number of edits within a reasonable range. Exceeding this limit can lead to model collapse. We introduce item of **editing endurance** to quantify this limit. Please refer to **general response** part for more detailed explanation or experiments.
>
> > Q4: There are also several different ... as well.
>
> **Reply:** Thanks for comment. Here are some clarifications:
>
> **Part A: Other dimensions for analysis.**
>
> (1) **Scale of the model:** This factor is actually the RQ3 of our paper, with detailed analysis and experiments presented in Section 4.3.
>
> (2) **Prompting conditions:** We adopted widely used prompt settings and hyperparameters from existing literature and technical reports (e.g., CoT usage, number of few-shot prompting). These details are fully listed in Appendix D.1.
>
> (3) **Whether models benefit from RAGs or not:** This paper does not involve the use of RAGs.
>
> **Part B: Fairness and reliability of our evaluation**
>
> To ensure a fair and comprehensive evaluation, we utilize widely adopted commonly used benchmarks like MMLU and BBH. We adopt standard prompt settings and hyperparameters from existing literature and technical reports (e.g., CoT usage, few-shot prompting) and utilize inference acceleration frameworks (vLLM) to align with practical application. These details are thoroughly documented in Appendix D.1, ensuring fairness and reproducibility in evaluation.
>
> > Q5: In addition ... just API access.
>
> **Reply:** We will answer your question from two aspects:
>
> **(1) Why performance degradation occurs:**
>
> We believe that the performance degradation during sequential editing stems from the disruption of the LLM's "intrinsic knowledge structure." Specifically, the model's parameter distribution undergoes changes. With a limited number of edits, the parameter distribution shift is minimal, allowing for knowledge updates while preserving performance. However, extensive editing leads to significant changes in the parameter distribution, disrupting the knowledge encoded within the parameters and resulting in performance decline.
>
> **(2) Detecting performance degradation with different access levels:**
>
> **API Access:** Users with API access cannot directly edit the model's knowledge. However, they can assess potential performance degradation by querying the model and evaluating its responses.
>
> **Full Weight Access:** Users with access to model weights can perform all operations  described in our paper.
>
> > Q6: In addition ... nor comprehensive.
>
> **Reply:** Thank you for your advice. We will add our new finding and analysis in the revision.

---

> ### Author Response · Authors · 2024-08-13
> **Kindly Reminder**
>
> Dear Reviewer 2vUP:
>
> Thank you very much for your valuable comments. In the rebuttal period, we have provided detailed responses to all your comments and questions point-by-point.
>
> A brief summary of our responses includes:
>
> + Q1: Can our findings generalize to other language models or other application scenarios.
> + Q2: Distinction between knowledge editing and FT
> + Q3: Deeper understanding and systematic analysis:  Elasticity and Plasticity Trade-off
> + Q4: Fairness and reliability of our evaluation & Other dimensions for analysis.
> + Q5: Why performance degradation occurs & Detecting performance degradation with different access levels.
>
> Please let us know if any remaining issues or further clarifications are needed. We are more than happy to provide additional information or explanations to help resolve any concerns.
>
> Thank you for your time and valuable feedback.
>
> Best regards

---

### Author Rebuttal · Authors · 2024-08-05

## **General Response to All of Reviewers**

We appreciate all the reviewers for their thoughtful comments and suggestions on our paper.

We are very glad to see that the reviewers find our focused problem is important and useful (R1, R2, R3) ,the insights are valueable and reliable (R1, R2) and the experiments are comprehensive and organized (R1, R2, R3). We are pleased that the reviewers find our writing is very clear and easy to understand (R2, R3).
We have tried our best to address the reviewers' comments and concerns in individual responses to each reviewer. The reviews allowed us to improve our draft.

In the following part,  we would like to provide more detailed version of important questions:

> Q1: Motivation and goal of our work

**Reply:** The primary objective of this paper is to provide a comprehensive evaluation of the impact of existing knowledge editing methods on LLM performance within the context of sequential editing. We aim to investigate potential factors influencing performance, offering valuable insights and guidance for future research in this area. Our focus is **not** on proposing specific new methods.

> Q2: Deeper understanding and analyis of our findings

**Reply:** We have re-organized the findings of our paper and made the following analysis and observation , which we term the Elasticity and Plasticity Trade-off (EPT):

First, we define two crucial concepts in sequential knowledge editing: plasticity, referring to the model's ability to retain newly acquired knowledge through editing, and elasticity, denoting the model's ability to retain its original knowledge after editing. Prior to editing, the model's knowledge solely comprises its inherent, intrinsic knowledge. Post-editing, the model's knowledge encompasses two components: updated knowledge and inherent knowledge.

We observe that during large-scale sequential editing, simultaneously maintaining high plasticity (retention of updated knowledge) and high elasticity (retention of inherent knowledge) is challenging. These two objectives are inherently in conflict. We refer to this phenomenon as the Elasticity and Plasticity Trade-off, drawing parallels to the Stability-Plasticity Dilemma observed in catastrophic forgetting during neural network fine-tuning. Existing work primarily focuses on the detrimental effect of large-scale sequential editing on elasticity, neglecting the interplay between elasticity and plasticity. Our work highlights this crucial trade-off, emphasizing the inherent conflict between these two objectives.

Experimental results with different editing methods on Llama2-7B (see Table 1,2,3) demonstrate that when applying editing, maintaining a balance between elasticity and plasticity requires limiting the number of edits within a reasonable range. Exceeding it can lead to model collapse, characterized by the destruction of the model's intrinsic knowledge structure.

 Results  in Table 1-3 also show that some methods can maintain the EPT over thousands of edits. However,  this capability is not unlimited. With a sufficiently large number of edits, model collapse still occurs. We introduce the concept of  **editing endurance** to quantify the maximum number of edits a model can undergo while preserving the EPT.

| # Edits |   MMLU |    BBH |  TrivialQA | Efficacy | Generalization | Locality |
| :------ | :----- | :----- | :--------- | :------- | :------------- | :------- |
| 0       | 0.459  | 0.400  | 0.525      | 17.4     | 19.2           | 86.1     |
| 10      | 0.457  | 0.396  | 0.524      | 100      | 94.4           | 92.2     |
| 100     | 0.443  | 0.377  | 0.212      | 100      | 94.8           | 91.8     |
| 500     | 0      | 0      | 0          | 98.2     | 92.5           | 80.7     |
| 1000    | 0      | 0      | 0          | 98.4     | 92.3           | 80.4     |

**Table 1: Edit result for MEMIT**

| # Edits | MMLU  | BBH   | TrivialQA | Efficacy | Generalization | Locality |
| :------ | :---- | :---- | :-------- | :------- | :------------- | :------- |
| 0       | 0.459 | 0.400 | 0.525     | 17.4     | 19.2           | 86.1     |
| 10      | 0.459 | 0.401 | 0.523     | 100      | 95.6           | 81.3     |
| 100     | 0.459 | 0.396 | 0.521     | 100      | 95.5           | 81.6     |
| 500     | 0.456 | 0.392 | 0.499     | 99.4     | 95.4           | 80.7     |
| 1000    | 0.457 | 0.392 | 0.490     | 99.1     | 95.4           | 80.2     |

**Table 2: Edit result for PMET**

| # Edits | MMLU  | BBH   | TrivialQA | Efficacy | Generalization | Locality |
| :------ | :---- | :---- | :-------- | :------- | :------------- | :------- |
| 0       | 0.459 | 0.400 | 0.525     | 17.4     | 19.2           | 86.1     |
| 10      | 0.457 | 0.394 | 0.528     | 99.6     | 51.2           | 54.2     |
| 100     | 0.458 | 0.394 | 0.527     | 92.4     | 47.3           | 54.7     |
| 500     | 0.457 | 0.399 | 0.528     | 86.4     | 42.4           | 57.6     |
| 1000    | 0.457 | 0.397 | 0.528     | 68.3     | 46.5           | 53.8     |

**Table 3: Edit result for MEND**

To provide an intuitive understanding of the EPT, we examine the weight changes with different edits in edited layer (see Table 4). This analysis aims to demonstrate that maintaining both elasticity and plasticity necessitates keeping weight changes within a reasonable range like PMET (neither too big nor too small). MEND lose the plasticity, while MEMIT lose the elasticity.

| Method | 0       | 10      | 100     | 1000     |
| :----- | :------ | :------ | :------ | :------- |
| ROME   | 117.053 | 118.265 | 269.497 | overflow |
| MEMIT  | 116.579 | 116.911 | 121.681 | 9291.127 |
| PMET   | 116.579 | 116.558 | 116.667 | 117.749  |
| MEND   | 126.209 | 126.198 | 126.198 | 126.199  |

**Table 4: L2 Norm of edited layer for different edits on Llama2-7B model.**

----

**We appreciate your comments and time! We have tried our best to address your concerns.** Would you mind checking  and confirming if there are any unclear parts?

---

### Decision · Program_Chairs · 2024-09-25

**Decision:**

Accept (poster)

**Comment:**

This paper evaluates the impact of model editing on the general capabilities of LLMs. Specifically, the paper investigates various model editing methods with considering factors such as the number of edits, model scale, safety, and different aspects of the fundamental capabilities of edited models. Several interesting findings are provided by the experimental results and the in-depth analysis.

Most of the concerns from the reviewers are addressed in the reviewer-author discussiion with additional results provided by the authors. This paper focuses on the real problem of model editing, and the exploration of how editing affects the safety of models is particularly valuable. While the paper indeed have some room for improvement, e.g., it could be better if the authors include more analysis of the elasticity and plasticity trade-off and the generalization of the findings to other types of language models in the revision, I think this paper deserves to be presented. Therefore, I recommend an "acceptance".